# ProPINN: Demystifying Propagation Failures in Physics-Informed Neural Networks

**Yuezhou Ma**[*]                                    *mayuezhou20@gmail.com*
**Haixu Wu**[*]                                       *wuhaixu98@gmail.com*
**Hang Zhou**                                         *zhou-h23@mails.tsinghua.edu.cn*
**Huikun Weng**                                       *wenghk22@mails.tsinghua.edu.cn*
**Jianmin Wang**                                      *jimwang@tsinghua.edu.cn*
**Mingsheng Long**[†]                                 *mingsheng@tsinghua.edu.cn*

*School of Software, BNRist, Tsinghua University, Beijing 100084, China*
[*]*Equal contribution.* [†]*Corresponding author.*

**Reviewed on OpenReview:** *https://openreview.net/forum?id=Wy54lrFd46*

## Abstract

Physics-informed neural networks (PINNs) have earned high expectations in solving partial differential equations (PDEs), but their optimization usually faces thorny challenges due to the unique derivative-dependent loss function. By analyzing the loss distribution, previous research observed the *propagation failure* phenomenon of PINNs, intuitively described as the correct supervision for model outputs cannot "propagate" from initial states or boundaries to the interior domain. Going beyond intuitive understanding, this paper provides a formal and in-depth study of propagation failure and its root cause. Based on a detailed comparison with classical finite element methods, we ascribe the failure to the conventional single-point-processing architecture of PINNs and further prove that propagation failure is essentially caused by the lower *gradient correlation* of PINN models on nearby collocation points. Compared to superficial loss maps, this new perspective provides a more precise quantitative criterion to identify where and why PINN fails. The theoretical finding also inspires us to present a new PINN architecture, named ProPINN, which can effectively unite the gradients of region points for better propagation. ProPINN can reliably resolve PINN failure modes and significantly surpass advanced Transformer-based models with 46% relative promotion.

## 1 Introduction

Accurately solving physics equations is essential to both scientific and engineering domains (Wazwaz, 2002; Roubíček, 2013). However, it is usually hard to obtain the analytic solution of PDEs. Therefore, classical numerical methods (Šolín, 2005; Dhatt et al., 2012) have been widely explored and served as a foundation for modern engineering (Ames, 2014). Recently, deep models have empowered significant progress in various domains and have also been applied in solving PDEs (Wang et al., 2023a). As one pioneering progress, physics-informed neural networks (PINNs) are proposed and widely studied (Raissi et al., 2019; Hao et al., 2022), which can approximate PDE solutions by formalizing equation constraints, initial and boundary conditions as a loss function and enforcing the outputs and gradients of neural networks to satisfy target PDEs. Taking advantage of the automatic differentiation feature of deep learning frameworks (Bradbury et al., 2018; Paszke et al., 2019), PINN can accurately calculate the derivative terms for physical quantities in equations without domain discretization, posing a promising direction for solving PDEs.

Although PINNs have attracted great attention, they still face serious challenges in enabling robust training and can fail in some "simple" PDEs, which are called PINN failure modes (Krishnapriyan et al., 2021).

(a) Ground Truth    (b) Vanilla PINN Prediction Analysis    (c) ProPINN (Ours) Prediction Analysis

Figure 1: Comparison of PINN and ProPINN on Convection ($\frac{\partial u}{\partial t} + 50\frac{\partial u}{\partial x} = 0$). In addition to the error map and residual loss, we also plot the gradient correlation of corresponding models between nearby points, which is newly proposed to identify the propagation failure. A low gradient correlation value (the darker color) indicates that the area is hard to propagate. See Appendix A for more results.

Researchers have attempted to tackle the training difficulties with new sampling strategies (Wu et al., 2023; Daw et al., 2023), loss functions (Yu et al., 2022; Wu et al., 2024), optimizers (Rathore et al., 2024), automatic differential methods (Shi et al., 2024), etc. Especially, Daw et al. noticed a special failure phenomenon during PINN optimization: the interior domain solely optimized by equation constraints will likely converge to a trivial solution. As shown in Figure 1(b), the equation constraint loss of PINN is sufficiently small but the approximated solution is still far from the ground truth. They attributed this to the failure in propagating the supervision of correct solution value from initial states or boundaries to interior points, calling it *propagation failure*. Despite this concept seeming intuitive, the formal and rigorous definition of "propagation" processes during PINN optimization and the root cause of propagation failure are still underexplored.

Some works start from observable loss maps and propose new sampling strategies to accumulate collocation points in areas with high residual losses to break the propagation "barriers" (Daw et al., 2023; Anonymous, 2024). Although they provide practical remedial measures, these loss-oriented methods may not solve the propagation issue at its root. As illustrated in Figure 1(b), we can find that PINN fails at the beginning according to the actual error map, while the residual loss distribution is too dispersed to identify the beginning failure. To demystify the propagation failure of PINNs, we draw inspiration from traditional numerical methods, which rarely suffer from propagation issues. From the comparison with finite element methods (FEMs) (Dhatt et al., 2012), we realize that the primary cause of propagation failure lies in the conventional design principle of PINN model architectures. Unlike FEMs that discretize the input domain as connected meshes, PINNs usually treat the input domain as a set of independently processed collocation points, which makes the optimization of different positions relatively independent, thereby reducing the "interaction" among PINN outputs on nearby positions and resulting in poor propagation.

Based on the above analysis, this paper theoretically establishes that low gradient correlation between nearby collocation points characterizes weak local propagation in PINNs. Going beyond the residual loss mainly focused on by previous research (Wu et al., 2023; Daw et al., 2023; Zhao et al., 2024), gradient correlation provides a foundational understanding of propagation failure, which can serve as a precise criterion to identify propagation issues. For example, in Figure 1(b), the area with the lowest gradient correlation corresponds well to the zone that first appears to have high error. With the idea of enhancing gradient correlation, we present ProPINN as a simple but effective PINN architecture, which can efficiently unite region gradients to boost propagation. ProPINN successfully mitigates the propagation failure and achieves the consistent state-of-the-art performance in various PDEs, surpassing advanced Transformer-based models. Overall, our contributions can be summarized as follows:

- Based on a detailed comparison with FEMs, we initially define the *propagation failure* from the model architecture perspective and prove that the root cause of failure is low gradient correlation, which can serve as a precise criterion to identify PINN failures.

- ProPINN with multi-region mixing mechanism is presented as an efficient architecture, which can tightly unite the optimization of collocation points within a region and improve the gradient correlation of nearby points for tackling propagation failure.

- ProPINN can reliably mitigate PINN failure modes and achieve state-of-the-art with 46% relative gain in typical PDE-solving tasks with favorable efficiency.

## 2 Preliminaries

A PDE in $\Omega \subseteq \mathbb{R}^{d+1}$ with equations $\mathcal{F}$, initial and boundary conditions $\mathcal{I}, \mathcal{B}$ writes as:

$$\mathcal{F}(u)(\mathbf{x}) = 0, \mathbf{x} \in \Omega; \ \mathcal{I}(u)(\mathbf{x}) = 0, \mathbf{x} \in \Omega_0; \ \mathcal{B}(u)(\mathbf{x}) = 0, \mathbf{x} \in \partial\Omega. \tag{1}$$

Here $\mathbf{x} \in \Omega \subseteq \mathbb{R}^{d+1}$ denotes the position information of input points and $u : \mathbb{R}^{d+1} \to \mathbb{R}^m$ represents the target PDE solution (Wazwaz, 2002; Evans, 2010). Usually, $\mathbf{x} = (x_1, \cdots, x_d, t)$ contains both spatial and temporal position information and $\Omega_0$ correspond to the $t = 0$ situation. A physics-informed neural network $u_\theta$ will approximate the PDE solution $u$ by optimizing the following loss function (Raissi et al., 2019):

$$\mathcal{L}(u_\theta) = \frac{\lambda_{\text{res}}}{n_{\text{res}}} \sum_{i=1}^{n_{\text{res}}} \|\mathcal{F}(u_\theta)(\mathbf{x}_{\text{res}}^i)\|^2 + \frac{\lambda_{\text{ic}}}{n_{\text{ic}}} \sum_{i=1}^{n_{\text{ic}}} \|\mathcal{I}(u_\theta)(\mathbf{x}_{\text{ic}}^i)\|^2 + \frac{\lambda_{\text{bc}}}{n_{\text{bc}}} \sum_{i=1}^{n_{\text{bc}}} \|\mathcal{B}(u_\theta)(\mathbf{x}_{\text{bc}}^i)\|^2, \tag{2}$$

where $\lambda_*$ and $n_*$ represent the loss weights and the numbers of collocation points respectively.

**Propagation failures** Unlike the conventional supervised learning that directly constrains the model output, the residual loss ($\mathcal{F}$ item in Equation 2) only describes the derivative relation on different positions in $\Omega$ (e.g. $\frac{\partial u_\theta}{\partial x_i}, \frac{\partial u_\theta}{\partial t}$) and the direct supervision for model outputs $u_\theta(\mathbf{x})$ only exists on initial state $\Omega_0$ or boundaries $\partial\Omega$. For interior points, only constraining model's gradients without any supervision for the model output may lead to a trivial solution. For example, without considering initial and boundary conditions, the all-zero function $u_\theta = 0$ is also a solution for the convection equation $\frac{\partial u_\theta}{\partial t} + 50 \frac{\partial u_\theta}{\partial x} = 0$. Thus, *propagation failure* is proposed by Daw et al.. Their key idea is that to obtain a correct solution in the whole domain, the correct supervision of model outputs must propagate from the initial or boundary points to the interior domain during training. Although Daw et al. provided an intuitive description of why propagation fails in PINN by analyzing the loss distribution, a formal and in-depth understanding of the root cause of propagation failure is still underexplored, which is formally proved in our paper from the model architecture perspective.

**Training strategies** To tackle optimization challenges of PINNs, training strategies have been widely explored, which can be roughly categorized into the following two branches.

The first branch focuses on sampling strategies to calibrate collocation points at each iteration. These works mainly focus on the areas with high residual loss (Krishnapriyan et al., 2021; Wang et al., 2022a; Wu et al., 2023; Anonymous, 2024). Especially, to mitigate the propagation failure described above, R3 (Daw et al., 2023) is proposed by accumulating sampled collocation points around the high-residual area to break propagation "barriers". However, all these methods primarily attempt to remedy propagation failure by sampling points, overlooking the inherent deficiency of PINN architecture: independent optimization among sampled points, which is explored in depth and improved further by our work.

Besides, PINN loss contained multiple components (Equation 2), making loss reweighting essential. Wang et al. (Wang et al., 2022b) proposes to adjust $\lambda_*$ to balance the convergence rate of different loss components. Considering the temporal causality of PDEs, causal PINN (Wang et al., 2024b) is presented to increase the loss weights of points in the subsequent based on the accumulated residual of previous time steps. Unlike these methods, we focus on the model architecture, which is orthogonal to these loss-oriented works.

**Model architectures** Vanilla PINN (Raissi et al., 2019) is essentially a multilayer perceptron (MLP). Afterward, QRes (Bu & Karpatne, 2021) and FLS (Wong et al., 2022) enhance the model capacity and position embedding respectively. Further, PirateNet (Wang et al., 2024a) leverages residual networks for better scalability. However, all of these methods still process collocation points independently, overlooking spatiotemporal correlations of PDEs. Recently, PINNsFormer (Zhao et al., 2024) first introduced Transformer (Vaswani et al., 2017) to PINNs and adopted the attention mechanism to capture temporal correlations among different points. Subsequently, SetPINN (Nagda et al., 2024) extends Transformer to a general spatiotemporal framework. Unlike these models, our proposed ProPINN stems from the in-depth study of propagation failures without relying on the computation-intensive Transformers, achieving better performance and efficiency.

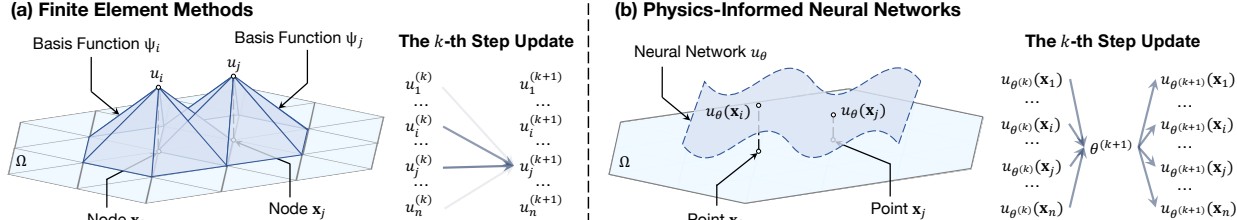

Figure 2: Comparison between (a) FEMs and (b) PINNs. Blue arrows highlight quantities with direct interactions during training. Compared to FEMs, solution values of PINNs among different positions are only under an implicit correlation by updating the model parameter $\theta$ during training.

## 3 Method

As aforementioned, we focus on the *propagation failure* of PINNs, which is considered as one of the foundation problems of PINN optimization. This section will first discuss the propagation properties of PINNs, where we take insights from FEMs to give an intrinsic understanding of why propagation failures exist. Based on theoretical results, we present ProPINN as a simple but effective PINN architecture, which achieves favorable propagation by uniting region gradients. All the proofs can be found in Appendix B.

### 3.1 Demystify Propagation Failure

Previous work (Daw et al., 2023) attributed the propagation failure to "some collocation points start converging to trivial solutions before the correct solution from initial/boundary points is able to reach them." However, we find that, in FEMs, the iteration will start from a trivial estimation and the interior areas also hold incorrect output supervision in the beginning iterations, while they are not affected by propagation issues. Thus, we believe there exists an unexplored root cause for propagation failures beyond these intuitive understandings.

To demystify propagation failures of PINNs, we make a detailed comparison between FEMs and PINNs on their parameter updating processes. As shown in Figure 2, parameters of FEMs are defined on discretized mesh points and nearby points directly affect each other during optimization, which is formally stated below.

**Theorem 3.1** (**Propagation in FEMs**). *(Dhatt et al., 2012) Suppose that FEMs discretize $\Omega$ into computation meshes with $n$ nodes $\{\mathbf{x}_i\}_{i=1}^n$ and approximate the PDE solution by optimizing coefficients of basis functions $\{\Psi_i\}_{i=1}^n$, which are defined as region linear interpolation. Denote the coefficient of basis function $\Psi_i$ as $u_i$, which is also the solution value of the i-th node. With the Jacobi iterative method for solution value update, the interaction among solution values $\{u_i\}_{i=1}^n$ at the k-th step is:*

$$u_j^{(k+1)} = \frac{1}{D(\Psi_j, \Psi_j)} \left( b_j - \sum_{i \neq j} D(\Psi_i, \Psi_j) u_i^{(k)} \right), \tag{3}$$

*where $\{b_j\}_{j=1}^n$ are constants related to external force. $D(\cdot, \cdot)$ is a variational version of PDE equation $\mathcal{F}(\cdot)$, which presents non-zero values only for overlapped basis functions.*

**Remark 3.2** (**FEMs are under active propagation**). *Theorem 3.1 indicates that coefficients of basis functions with overlap area are explicitly correlated. Thus, $u_i^{(k)}$'s change will directly affect the values of its adjacent nodes, ensuring an active propagation for the entire domain.*

**Remark 3.3** (**Physical meanings of Equation 3**). *The parameter update in FEMs relies on $D(\cdot, \cdot)$, which is also named as stiffness matrix in solid mechanics (Yang & McGuire, 1986). $D(\Psi_i, \Psi_j)$ describes the force on the j-th node to make region balance when the i-th node has a unit displacement.*

Inspired by analyses of FEMs, we define the propagation in PINNs in terms of the influence of each point's value change on other points, yielding the first formal measurement of the propagation failure. Since in PINNs, the model output is determined by model parameter $\theta$, we propose to represent "value change" from

the gradient perspective, namely $\frac{\partial u_\theta}{\partial \theta}\big|_{\mathbf{x}}$. This perspective help us redefine the stiffness matrix from FEMs (Yang & McGuire, 1986) in the context of PINN models, bridging the computation gap between PINNs and FEMs. Specifically, the propagation failure of PINNs can be formally defined as follows.

**Definition 3.4** (**Propagation failure in PINN**). *In spirit of the physics meaning of Equation 3, we define the "stiffness" coefficient between $\mathbf{x}$ and $\mathbf{x}'$ for PINN $u_\theta$ as the "slope" w.r.t. the parameter change:*

$$D_{PINN}(\mathbf{x}, \mathbf{x}') = \lim_{\lambda \to 0} \frac{\left\| u_\theta(\mathbf{x}') - u_{\theta - \lambda \frac{\partial u_\theta}{\partial \theta}\big|_{\mathbf{x}}}(\mathbf{x}') \right\|}{\lambda}, \tag{4}$$

*which measures the impact on model output at $\mathbf{x}'$ after updating PINN with an standard gradient step at $\mathbf{x}$. This formula is analogous to applying a unit force at $\mathbf{x}$ and observing a displacement at $\mathbf{x}'$. If $\mathbf{x}$ and $\mathbf{x}'$ are adjacent and $D_{PINN}(\mathbf{x}, \mathbf{x}')$ is less than an empirically defined threshold $\epsilon$, we term propagation failure occurs between them.*

**Remark 3.5** (**Region propagation**). *In FEMs shown in Figure 2(a), only overlapped basis functions (corresponding to adjacent discretization points) could have a non-zero stiffness coefficient. Thus, we only discuss propagation among nearby points in Equation 4.*

Based on the above formal definition, we can further derive the root cause for propagation failures, which transforms the physical-meaning-based definition into a deep model property.

**Theorem 3.6** (**Gradient-correlation**). *For a PINN $u_\theta$ and adjacent points $\mathbf{x}, \mathbf{x}' \in \Omega$, under the definition in Definition 3.4, the PINN stiffness coefficient is equivalent to the gradient correlation between two points:*

$$D_{\text{PINN}}(\mathbf{x}, \mathbf{x}') = G_{u_\theta}(\mathbf{x}, \mathbf{x}') = \left\| \left\langle \frac{\partial u_\theta}{\partial \theta}\bigg|_{\mathbf{x}}, \frac{\partial u_\theta}{\partial \theta}\bigg|_{\mathbf{x}'} \right\rangle \right\|. \tag{5}$$

*Therefore, weak propagation between adjacent points, as measured by a small $D_{\text{PINN}}(\mathbf{x}, \mathbf{x}')$, is equivalently characterized by a small gradient correlation $G_{u_\theta}(\mathbf{x}, \mathbf{x}')$.*

**Remark 3.7** (**Optimizer-agnostic interpretation**). *The standard gradient step in Definition 3.4 is used only to expose the propagation stiffness analytically. Although optimizers such as Adam and L-BFGS modify raw gradients, their updates are still gradient-induced. Thus,* weak gradient correlation *between nearby collocation points still indicates* weak local propagation *under gradient-based optimization.*

**Why propagation failures exist in PINNs** According to the definition of Equation 5, the gradient correlation is an inner product of high-dimensional tensors, where $\frac{\partial u_\theta}{\partial \theta}\big|_* \in \mathbb{R}^{m \times |\theta|}$, $|\theta|$ denotes the parameter size (usually $> 10^3$) and $m$ is the output dimension. Thus, these gradient tensors are easy to be orthogonal in the high-dimensional space (Ball, 1997), especially when different positions are independently optimized. For example, position $\mathbf{x}$ and position $\mathbf{x}'$ could correspond to different parts of model parameters. Further, this analysis also provides insights for "why PINNs cannot benefit from large models" (Wang et al., 2024a), since a larger parameter size $|\theta|$ is more likely to cause orthogonal gradients.

### 3.2 Propagation Physics-Informed Neural Networks

Theorem 3.6 highlights that the gradient correlation is the foundation item that affects the propagation of PINNs. Note that the gradient correlation is essentially determined by the model architecture. Thus, we present ProPINN as a new PINN architecture to enhance the gradient correlation by introducing unified region gradient to each collocation point, which can effectively boost the propagation. As shown in Figure 3, the key component of ProPINN is a multi-region mixing mechanism, which is a well-thought-out design considering both efficiency and performance. The following are the details.

**Differential perturbation** Given a single collocation point $\mathbf{x} \in \Omega \subseteq \mathbb{R}^{(d+1)}$, we first augment it by perturbing its position within multiscale regions. Note that different from PINNsFormer (Zhao et al., 2024) and SetPINN (Nagda et al., 2024) that only utilize augmented representations of multiple points but detach

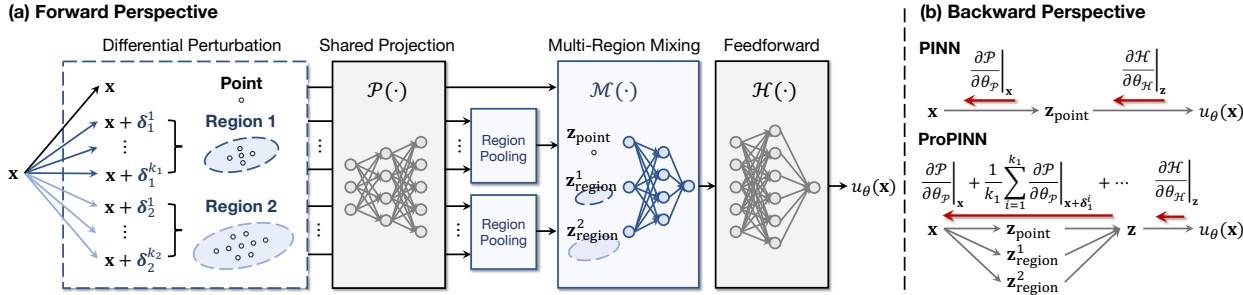

Figure 3: ProPINN in both (a) forward and (b) backward perspectives, where the single input point is augmented to point sets in multiscale regions, which can efficiently promote interaction between model parameter gradients on different positions within multiple regions.

the gradient backpropagation, ProPINN leverages the perturbation as a differential layer and projects all the augmented points into deep representations with a shared projector $\mathcal{P}(\cdot)$, which is:

$$
\text{Diff-Aug}(\mathbf{x}) = \left\{ \mathbf{x}, \left\{ \{\mathbf{x} + \boldsymbol{\delta}_r^i\}_{i=1}^{k_r} \right\}_{r=1}^{\#\text{scale}} \right\},
$$

$$
\mathbf{z}_{\text{point}} = \mathcal{P}(\mathbf{x}), \left\{ \mathbf{z}_{\text{region}}^{i,r} \right\}_{i=1}^{k_r} = \left\{ \mathcal{P}(\mathbf{x} + \boldsymbol{\delta}_r^i) \right\}_{i=1}^{k_r}
\tag{6}
$$

where $\{\boldsymbol{\delta}_r^i\}_{i=1}^{k_r}$ are random perturbations for $\mathbf{x}$ within the $r$-th region whose size is $[-R_r, R_r]^{d+1}$, and $k_r$ is the corresponding number of perturbations. We denote the number of scales, i.e., the number of multiscale regions, as $\#\text{scale}$. $\mathbf{z}_{\text{point}}, \mathbf{z}_{\text{region}}^{i,r} \in \mathbb{R}^{d_{\text{model}}}$ are representations of the original point $\mathbf{x}$ and its multi-region augmentation $(\mathbf{x} + \boldsymbol{\delta}_r^i)$ respectively. $\mathcal{P} : \mathbb{R}^{d+1} \to \mathbb{R}^{d_{\text{model}}}$ is a lightweight MLP to encode the coordinates of collocation and augmented points. Notably, the design in considering different-scale regions not only covers multiscale properties of PDEs but also simulates non-uniform meshes in FEMs, where each point can selectively aggregate information from multiple regions.

From the forward perspective, the above design can naturally augment the receptive field. Moreover, from the backward perspective shown in Figure 3(b), the differential perturbation design can also aggregate gradients of collocation points within multiscale regions, thereby explicitly enhancing their gradient correlation.

**Remark 3.8** (**Connection to adversarial training**). *Although ProPINN also perturbs input coordinates, its motivation differs from adversarial training. Adversarial perturbations (Cohen et al., 2019) usually assume that nearby perturbed samples preserve the same label, whereas in PINNs each spatiotemporal coordinate has its own physical supervision from PDE residuals, boundary conditions, or initial conditions. Thus, ProPINN's perturbation is better viewed as representation-level local augmentation. Its differential design aims to unite nearby region gradients and enhance local propagation, rather than enforce robustness to input perturbations.*

**Multi-region mixing** After shared projection, we can obtain $(1 + \sum_{r=1}^{\#\text{scale}} k_r)$ representations from multi-region augmented positions. Previous Transformer-based studies, such as PINNsFormer (Nagda et al., 2024) and SetPINN (Nagda et al., 2024), apply the attention mechanism (Vaswani et al., 2017) among collocation points to capture complex spatiotemporal dependencies. However, since attention involves inner products among representations, it will also bring huge computation costs in both forward and backpropagation processes, especially for PINNs that usually need to calculate high-order gradients.

Instead of directly modeling dependencies among collocation points, we propose an efficient multi-region mixing mechanism. It first averages the representations in various regions to generate multiple region representations. Owing to the special property of PDEs, the pooled multiscale representations are still under the same PDE but with different coefficients (Graham et al., 2007). Thus, compared to complex dependencies among different positions, the relation among different coefficient PDEs is much more steady, allowing us to

Table 1: Summary of benchmarks, covering both standard PDE-solving tasks and complex fluid dynamics. #Dim denotes the dimension of input domain, "+T" refers to "time-dependent". We also visualize target solutions in (a) Karman Vortex and (b) Fluid Dynamics tasks, which involve complex spatiotemporal dynamics and multi-physics (velocity and pressure) interactions in the 2D+T space.

| Type | #Dim | Benchmarks | Property |
|------|------|-----------|----------|
| Standard Tasks | 1D+T | Convection 1D-Reaction Allen-Cahn | Failure Modes (2021) |
| | | 1D-Wave | High-order |
| Complex Fluid | 2D+T | Karman Vortex Fluid Dynamics | Navier-Stokes |

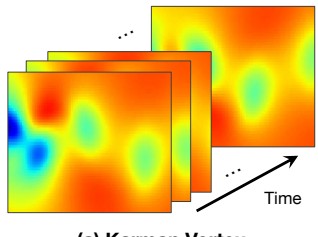

**(a) Karman Vortex**

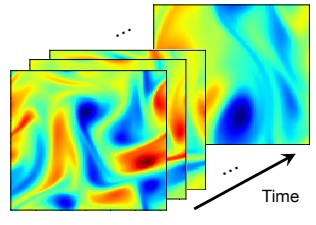

**(b) Fluid Dynamics**

adopt a simple linear mixing layer rather than the attention mechanism:

$$\mathbf{z}_{\text{region}}^r = \text{Pooling}\left(\left\{\mathbf{z}_{\text{region}}^{i,r}\right\}_{i=1}^{k_r}\right), r = 1, \cdots, \#\text{scale}$$

$$\mathbf{z} = \mathcal{M}\left(\mathbf{z}_{\text{point}}, \mathbf{z}_{\text{region}}^1, \cdots, \mathbf{z}_{\text{region}}^{\#\text{scale}}\right), \tag{7}$$

where $\mathcal{M} : \mathbb{R}^{(1+\#\text{scale}) \times d_{\text{model}}} \to \mathbb{R}^{d_{\text{model}}}$ is an MLP layer for mixing multi-region representations. Afterward, the mixed representation $\mathbf{z}$ is projected to the target dimension by another MLP layer $\mathcal{H} : \mathbb{R}^{d_{\text{model}}} \to \mathbb{R}^m$ for the final solution, namely $u_\theta(\mathbf{x}) = \mathcal{H}(\mathbf{z}) \in \mathbb{R}^m$.

**Gradient analysis** As presented in the visualization of Figure 1 (c), by uniting region gradients, ProPINN can successfully boost the gradient correlation within multiple regions for better propagation, which can also be theoretically understood through the following theorem.

**Assumption 3.9 (Correlation among region gradients).** *Given PINN $u_\theta$, we assume that there exists a region size $R > 0$, s.t. $\forall \mathbf{x}, \mathbf{x}' \in \Omega$ with $\|\mathbf{x} - \mathbf{x}'\| \le R$, $\left\langle \frac{\partial u_\theta}{\partial \theta}\big|_{\mathbf{x}}, \frac{\partial u_\theta}{\partial \theta}\big|_{\mathbf{x}'} \right\rangle \ge 0$.*

**Theorem 3.10 (Gradient correlation improvement).** *Under Assumption 3.9 with region size $R$, given $k$ perturbations $\{\boldsymbol{\delta}_i\}_{i=1}^k$ with $\|\boldsymbol{\delta}_i\| \le \frac{R}{3}$ and defining $u_\theta^{region}(\mathbf{x}) = u_\theta(\mathbf{x}) + \frac{\sum_{i=1}^k u_\theta(\mathbf{x}+\boldsymbol{\delta}_i)}{k}$, then $\forall \mathbf{x}, \mathbf{x}' \in \Omega$, if $\|\mathbf{x} - \mathbf{x}'\| \le \frac{R}{3}$, we have $G_{u_\theta}(\mathbf{x}, \mathbf{x}') \le G_{u_\theta^{region}}(\mathbf{x}, \mathbf{x}')$.*

**Remark 3.11 (Efficient design in ProPINN).** *Theorem 3.10 demonstrates that aggregating region points by differential perturbation (Equation 6) can enhance the gradient correlation of nearby points. Considering the efficiency, ProPINN limits the region aggregation only within the lightweight projection layer $\mathcal{P}$ instead of the entire model, which can effectively avoid propagation failure and is much more computation efficient.*

**Efficiency analysis** Compared to single-point-processing architectures, the extra computation of ProPINN comes from augmentation in Equation 6. Supposed that flops of $\mathcal{P}(\cdot)$ is $\text{ops}(\mathcal{P})$, the extra cost of ProPINN is $\sum_{r=1}^{\#\text{scale}} k_r \text{ops}(\mathcal{P})$. Although this seems like a huge overload, it will not affect the practical efficiency of ProPINN significantly. This efficiency benefits from the parallel computation in both forward and backward computation graphs (Paszke et al., 2019), as well as the lightweight design of the projection layer ($\mathcal{P}$). In our experiments, ProPINN is 2-3× faster than other explicit dependency modeling methods (e.g. PINNsFormer (Zhao et al., 2024) and SetPINN (Nagda et al., 2024)) and comparable with other single-point-processing PINNs (e.g. QRes (Bu & Karpatne, 2021) and FLS (Wong et al., 2022)) but with 60%+ relative promotion.

## 4 Experiments

We widely test ProPINN in extensive PDEs and provide detailed comparisons with advanced PINN architectures in performance, efficiency and scalability. All the implementation details can be found in Appendix C.

Table 2: Performance comparison of different PINN architectures on standard PDE-solving tasks, which usually appear failure modes (Krishnapriyan et al., 2021). Both rMAE and rRMSE are recorded. Smaller values indicate better performance. For clarity, the best result is in bold and the second best is underlined. Promotion refers to the relative error reduction w.r.t. the second best model ($1 - \frac{\text{Our error}}{\text{The second best error}}$).

| Model | Convection | | 1D-Reaction | | Allen-Cahn | | 1D-Wave | |
|---|---|---|---|---|---|---|---|---|
| | rMAE | rRMSE | rMAE | rRMSE | rMAE | rRMSE | rMAE | rRMSE |
| Vanilla PINN (Raissi et al., 2019) | 0.778 | 0.840 | 0.982 | 0.981 | 0.350 | 0.562 | 0.326 | 0.335 |
| QRes (Bu & Karpatne, 2021) | 0.746 | 0.816 | 0.979 | 0.977 | 0.942 | 0.946 | 0.523 | 0.515 |
| FLS (Wong et al., 2022) | 0.674 | 0.771 | 0.984 | 0.985 | 0.357 | 0.574 | 0.102 | 0.119 |
| KAN (Liu et al., 2024) | 0.922 | 0.954 | 0.031 | 0.061 | 0.352 | 0.563 | 0.499 | 0.489 |
| PirateNet (Wang et al., 2024a) | 1.169 | 1.287 | 0.017 | 0.044 | 0.098 | 0.179 | 0.051 | 0.055 |
| PINNsFormer (Zhao et al., 2024) | 0.023 | 0.027 | 0.015 | 0.030 | 0.331 | 0.529 | 0.270 | 0.283 |
| SetPINN (Nagda et al., 2024) | 0.028 | 0.033 | 0.018 | 0.034 | 0.381 | 0.601 | 0.347 | 0.353 |
| **ProPINN (Ours)** | **0.018** | **0.020** | **0.010** | **0.020** | **0.036** | **0.087** | **0.016** | **0.016** |
| Promotion | 22% | 26% | 33% | 33% | 63% | 51% | 69% | 71% |

**Benchmarks**  As listed in Table 1, we experiment with six PDE-solving tasks, covering diverse 1D and 2D time-dependent PDEs. Specifically, solutions of Convection, 1D-Reaction and Allen-Cahn contain some steep areas, which are challenging to approximate and have been used to demonstrate PINN failure modes (Krishnapriyan et al., 2021) and propagation failures (Daw et al., 2023). Besides, 1D-Wave involves second-order derivatives, making it hard to optimize. In addition to the above standard benchmarks, we also test ProPINN with extremely challenging fluid dynamics, which are governed by intricate Navier-Stokes equations (Constantin & Foiaş, 1988). *Karman Vortex* is from (Raissi et al., 2019), which describes the fluid dynamics around a cylinder, exhibiting the famous Karman vortex street (Wille, 1960). *Fluid Dynamics* is from (Wang et al., 2023b) and involves fast dynamics of fluid on a torus. More details can be found in Appendix C.1.

**Baselines**  In addition to vanilla PINN (Raissi et al., 2019), we also compare ProPINN with other six PINN architectures. QRes (Bu & Karpatne, 2021), FLS (Wong et al., 2022), KAN (Liu et al., 2024) and PirateNet (Wang et al., 2024a) are under the conventional PINN architecture, where different collocation points are independently optimized. PINNsFormer (Zhao et al., 2024) and SetPINN (Nagda et al., 2024) are based on the Transformer backbone to capture spatiotemporal correlations inside PDEs. PirateNet and PINNsFormer are previous state-of-the-art. In addition, we also integrate ProPINN with sampling strategy R3 (Daw et al., 2023), loss reweighting method (Wang et al., 2022b) and optimization algorithm RoPINN (Wu et al., 2024) to verify that these methods contribute orthogonally to us.

**Implementations**  For all benchmarks, we set the number of regions #scale = 3 with region size $\{R_1, R_2, R_3\} = \{10^{-2}, 5 \times 10^{-2}, 9 \times 10^{-2}\}$, number of perturbations $\{k_1, k_2, k_3\} = \{3^{(d+1)}, 5^{(d+1)}, 7^{(d+1)}\}$ and representation dimension $d_{\text{model}} = 32$. For Convection, 1D-Reaction, Allen-Cahn, 1D-Wave and Karman Vortex, we follow (Zhao et al., 2024) and train the model with L-BFGS optimizer (Liu & Nocedal, 1989) for 1,000 iterations in PyTorch (Paszke et al., 2019). As for Fluid Dynamics, we follow (Wang et al., 2023b) and experiment with JAX (Bradbury et al., 2018). Relative L1 Error (rMAE) and relative Root Mean Square Error (rRMSE) are recorded. See Appendix C for more implementation details.

### 4.1 Standard Benchmarks

**Main results**  From results presented in Table 2, we can obtain the following key observations.

*(i)* ProPINN successfully mitigates PINN failure modes in Convection, 1D-Reaction and Allen-Chan. Its outstanding performance in 1D-Wave also verifies its capability in handling high-order PDEs. Notably, ProPINN also beats the latest Transformer-based models PINNsFormer (Zhao et al., 2024) and SetPINN (Nagda et al., 2024) with 46% rMAE reduction averaged from four PDEs, highlighting the advantage of our method.

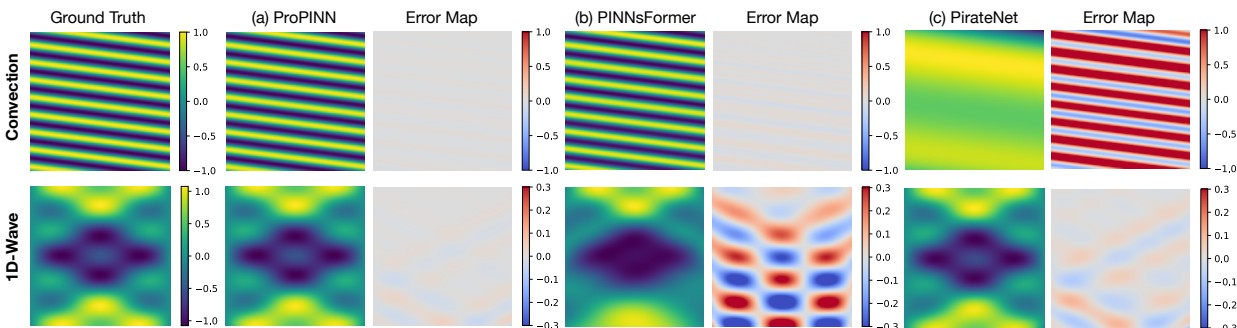

Figure 4: Visualization of model approximated solutions. Error map $(u_\theta - u)$ is also plotted.

*(ii)* It is also observed that only architectures that consider interactions among multiple points (PINNsFormer, SetPINN and ProPINN) consistently successfully converge and yield accurate results across all four tasks. All the models under the single-point-processing paradigm fail in Convection. This sharp dichotomy justifies our preceding theoretical discussion about the deficiency of single-point architectures, where independent optimization can easily cause inconsistent local gradient updates and thus propagation failure.

**Visualization**    To clearly compare model capacity in solving PDEs, we also visualize approximated solutions in Figure 4. We can find that PirateNet, under the single-point-processing paradigm, fails in handling the steep variations in Convection. As for the 1D-Wave with high-order derivatives, we find that PINNsFormer yields an insufficient performance, which may be because of the optimization difficulty of the attention mechanism under high-order loss. In sharp contrast, ProPINN demonstrates consistent and highly accurate performance in tackling both the rapid spatial changes in the Convection equation and the demanding optimization landscape of the 1D-Wave equation. This superior ability conclusively highlights the effectiveness and robustness of our proposed architecture in addressing a diverse range of PDE challenges.

**Efficiency comparison**    To verify the practicability of our method, we also provide the efficiency comparison in Figure 5. It is observed that ProPINN is about 2-3× faster than recent Transformer-based models: PINNsFormer (Zhao et al., 2024) and SetPINN (Nagda et al., 2024). Also, benefiting from our lightweight projection layer and parallel computing, ProPINN is comparable to single-point-processing PINNs in efficiency but brings more than 60%+ error reduction, achieving a favorable performance-efficiency trade-off.

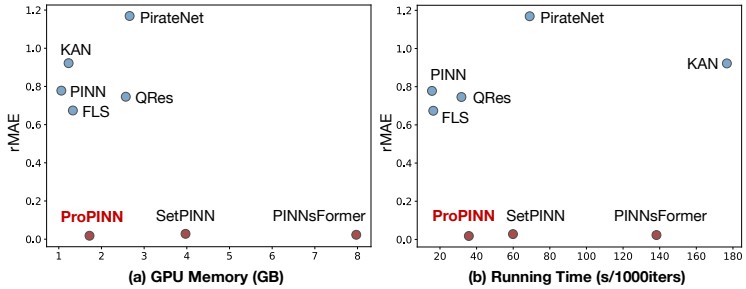

Figure 5: Efficiency comparisons on the Convection. Models under the single-point-processing paradigm are colored in blue, while models that consider point correlations are in red.

## 4.2   Complex Physics

As a long-standing mathematical problem, Navier-Stokes equations (Temam, 2001) for fluid dynamics have shown significant challenges and profound importance in real-world applications (Doering & Gibbon, 1995). Thus, in addition to standard benchmarks, we also experiment with Karman Vortex and Fluid Dynamics, which involve extremely intricate PDEs and spatiotemporal dynamics as shown in Table 1.

**Main results**    Compared to PDEs listed in Table 2, fluid simulation tasks in this section are much more complex. As shown in Table 3, ProPINN still achieves the best performance with over 30% relative promotion on average than the previous best model. Notably, ProPINN performs fairly well in the Fluid Dynamics task, which requires the model to accurately calculate velocity and pressure solutions for the whole spatiotemporal

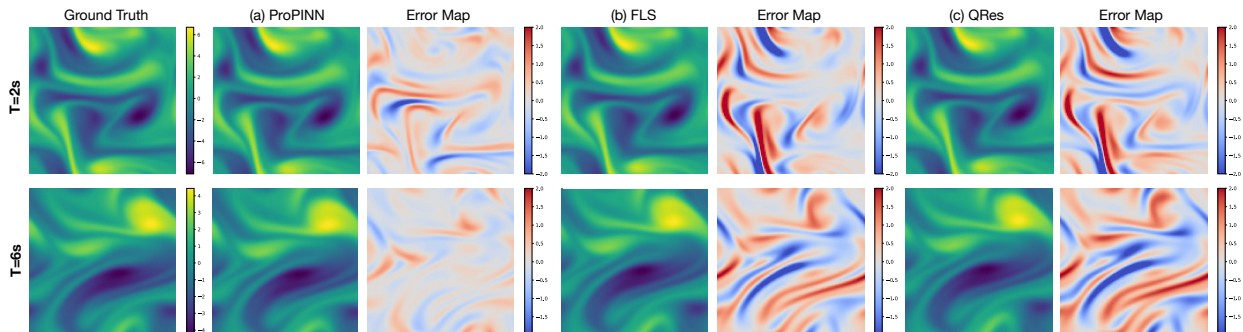

Figure 6: Visualization of the Fluid Dynamics task. Error map $(u_\theta - u)$ is also plotted.

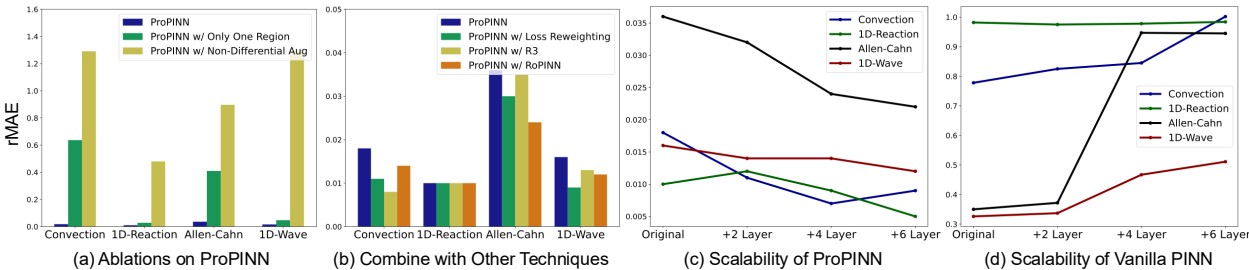

Figure 7: Model Analysis. (a) Ablations on multi-region mixing in Equation 7 and differential perturbation in Equation 6. (b) Integration with other techniques. (c-d) scaling ProPINN and PINN by adding layers.

sequence purely based on equation supervision. This task is under rapid and ever-changing dynamics and the last frame is far from its initial state (Table 1 (b)), making it extremely challenging. Besides, we also observe that PINNsFormer and SetPINN suffer from the training instability, which confirms our previous discussion on the training difficulties of the attention mechanism, further demonstrating the effectiveness of our MLP-based design in multi-region mixing.

**Visualization** As presented in Figure 6, ProPINN can accurately simulate the future variations of fluid, including the inner complex vortexes and distortions, even if the future frames are significantly changed. This indicates that ProPINN can give a precise solution for the Navier-Stokes equations, especially in processing the convection term, which is usually considered highly nonlinear and extremely complex.

Table 3: Comparison solving 2D time-dependent Navier-Stokes equations. "Nan" indicates that this model encounters the training instability problem. Since Fluid Dynamics is based on JAX (Bradbury et al., 2018) and it is hard to transfer the PyTorch implementation of KAN to JAX, we did not test it in this task, labeled in "/".

| Model | Karman Vortex | | Fluid Dynamics | |
|---|---|---|---|---|
| | rMAE | rRMSE | rMAE | rRMSE |
| Vanilla PINN (Raissi et al., 2019) | 13.08 | 9.08 | 0.3659 | 0.4082 |
| QRes (Bu & Karpatne, 2021) | 6.41 | 4.45 | 0.2668 | 0.3144 |
| FLS (Wong et al., 2022) | 3.98 | 2.77 | 0.2362 | 0.2765 |
| KAN (Liu et al., 2024) | 1.43 | 1.25 | / | / |
| PirateNet (Wang et al., 2024a) | 1.24 | 1.16 | 0.4550 | 0.5232 |
| PINNsformer (Zhao et al., 2024) | 0.384 | 0.280 | Nan | Nan |
| SetPINN (Nagda et al., 2024) | 0.287 | 0.209 | Nan | Nan |
| **ProPINN (Ours)** | **0.161** | **0.124** | **0.1834** | **0.2172** |
| Promotion | 44% | 41% | 22% | 21% |

### 4.3 Model Analysis

**Ablations** We provide a detailed ablation in Figure 7(a), including experiments with only one region in Equation 7, i.e. #scale = 1 and detaching gradients of augmented points in Equation 6. Results demonstrate that both multi-region mixing and differential perturbation are essential for the final performance. Especially, the non-differential perturbation will seriously damage the model's performance, even though it utilizes

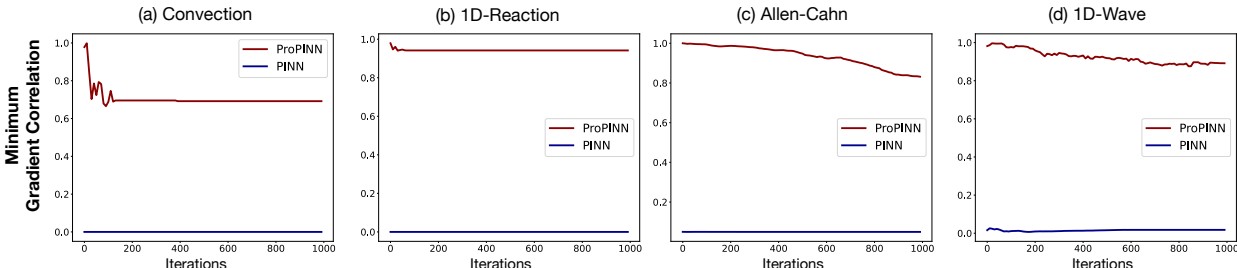

Figure 8: Visualization of minimum gradient correlation during training on all the standard benchmarks.

more collocation points to augment the receptive field. This finding further confirms that, compared with augmenting representation, uniting gradients of region points is more important for PINN optimization, verifying that region gradient correlations are the key factor.

**Integrating with other strategies**   As we discussed in related work, ProPINN mainly focuses on architecture design, which is orthogonal to previous research on training strategies. To verify their orthogonal contributions, we integrate ProPINN with the loss reweighting method (Wang et al., 2022b) and sampling strategy R3 (Daw et al., 2023) and optimization algorithm RoPINN (Wu et al., 2024). As illustrated in Figure 7(b), these methods can further boost ProPINN. As for 1D-Reaction, which is relatively simple, ProPINN's performance is nearly optimal; thus, the integration does not bring further promotion.

**Model scalability**   Theorem 3.6 attributes the propagation failure to lower region gradient correlations. As we discussed in Remark, since gradient correlation is defined as the inner product of high-dimensional gradient tensors in size $m \times |\theta|$, the correlation is easier to be orthogonal when adding model parameters. Thus, vanilla PINN presents a clear performance drop when scaling to larger models (Figure 7(d)). In contrast, ProPINN successfully mitigates propagation issues by introducing region gradients, which not only tackle PINN failure modes but also empower the model with favorable scalability.

**Analysis on Gradient Correlation**   As illustrated in the Figure 8, the change in gradient correlation during training reveals distinct behaviors for those architectures. For the standard PINN, we observe that the orthogonality of adjacent gradients progressively worsens throughout the training process. This phenomenon is likely due to the model becoming increasingly "overfitted" to individual adjacent points. Specifically, the model learns to deliver distinct, often conflicting, optimization directions to independently "overfit" two neighboring points, leading to a diminished gradient correlation. In contrast, the gradient correlation for ProPINN initially experiences a slight decrease but then stabilizes at a significantly high value (well above the predefined threshold $\epsilon$). This outcome clearly demonstrates the superiority of our proposed method in preserving information propagation across the multiple regions.

**Analysis on Training dynamics: training loss and test error**   As shown in Figure 9, the evolution of the training loss and test error diverges. The Vanilla PINN tends to prematurely "converge" to a low training loss, yet the corresponding test error remains relatively high. This behavior stems from the PINN's susceptibility to propagation failure, which effectively "blocks" the optimization process from further exploring the solution landscape. In contrast, ProPINN does not exhibit this limitation. Specifically on the Allen-Cahn dataset, ProPINN's effective design for propagation maintenance leads to a remarkable observation: as the training loss plateaus in the late stages of optimization, the test error continues to reduce significantly. *Note on training dynamics*: Given that this paper primarily focuses on model architecture design, unlike NTK-based methods (Jacot et al., 2018) which delve into the detailed training dynamics or convergence properties of PINNs, we reserve a more in-depth discussion on the convergence behavior for future work.

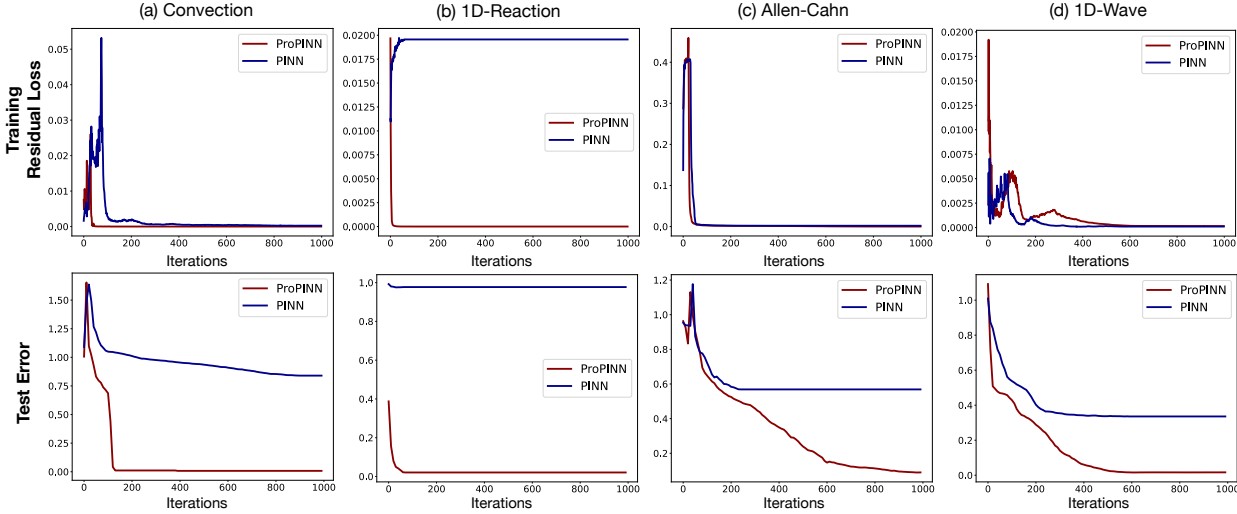

Figure 9: Training dynamics of ProPINN and PINN: training loss and test error on standard benchmarks.

## 5 Conclusion

This paper focuses on the propagation failures of PINNs and provides a formal and in-depth study of this crucial phenomenon. Going beyond the intuitive understanding, we theoretically proved that the root cause of propagation failures is the lower gradient correlation among nearby points, which can serve as a precise and quantifiable criterion for PINN failures. Inspired by the above theoretical analyses, ProPINN is presented as a new PINN architecture, which can effectively unite the gradients of region points for better information propagation. Experimentally, ProPINN can naturally enhance region gradient correlation and achieve remarkable promotion on standard benchmarks and challenging PDE-solving tasks with a favorable trade-off between performance and computational efficiency, concurrently presenting better scalability.

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

# A  Visualization of Gradient Correlation

As a supplement to Figure 1, we provide the visualization for the gradient correlation of all the other standard benchmarks here, where we can obtain the following observations:

- ProPINN generally shows higher gradient correlations than vanilla PINN in all the tasks, highlighting the effectiveness of multi-region mixing.

- In PINN, areas with small gradient correlations correspond perfectly to the boundary for higher error zones, indicating the effect of propagation failure.

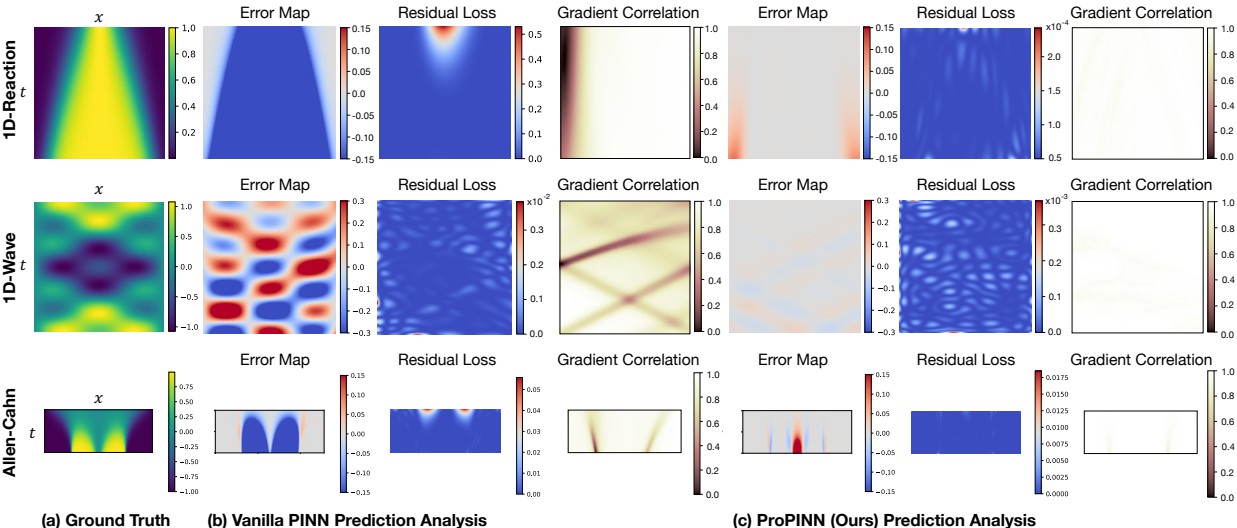

Figure 10: Visualization of gradient correlations on the other three standard benchmarks.

# B  Proof of Theorems in the Main Text

This section will present proofs for theorems in Section 3.

## B.1  Propagation in FEMs (Theorem 3.1)

This theorem is based on the general formalization of FEMs, which can be directly derived from textbook (Dhatt et al., 2012). Here, we reorganize the formalization to highlight the interaction among solution values of different areas in FEMs.

*Proof.* Without loss of generality, we consider the following PDE, which is defined in $\Omega \cup \partial\Omega$ with the following constraints:

$$\mathcal{F}(u)(\mathbf{x}) = f(\mathbf{x}), \mathbf{x} \in \Omega; \ u(\mathbf{x}) = 0, \mathbf{x} \in \partial\Omega, \tag{8}$$

where $f$ represents the function of external force.

Suppose that the solution $u \in \mathbb{U}$, where $\forall v \in \mathbb{U}, v|_{\partial\Omega} = 0$ and is with corresponding differential property to make the equation constraint $\mathcal{F}$ meaningful, then we can obtain the following variational formalization of PDE in Equation 8:

$$\int_{\Omega} (\mathcal{F}(u) - f)v\mathrm{d}x = 0, \ \forall v \in \mathbb{U}, \tag{9}$$

where $x$ refers to the variable of one dimension in $\Omega$.

Based on the integration by parts technique, it is easy to derive that

$$\int_\Omega (\mathcal{F}(u) - f)v\mathrm{d}x = \mathcal{F}'(u)v|_{\partial\Omega} - \int_\Omega \mathcal{F}'(u)\frac{\partial v}{\partial x}\mathrm{d}x - \int_\Omega fv\mathrm{d}x = -\int_\Omega \mathcal{F}'(u)\frac{\partial v}{\partial x}\mathrm{d}x - \int_\Omega fv\mathrm{d}x, \quad (10)$$

where $\frac{\partial\mathcal{F}'(u)}{\partial x} = \mathcal{F}(u)$. For clarity, we define $D(u, v) = \int_\Omega \mathcal{F}'(u)\frac{\partial v}{\partial x}\mathrm{d}x$ and $B(v) = -\int_\Omega fv\mathrm{d}x$. Thus, based on the above variational derivation, the PDE solving process is to find $u \in \mathbb{U}$ to satisfy the following equation:

$$D(u, v) - B(v) = 0, \forall v \in \mathbb{U}. \quad (11)$$

The key idea of FEM is to find an approximated solution on the computation mesh. Specifically, it is to find $\widehat{u} \in \widehat{\mathbb{U}}$ to satisfy the above-derived constraint, where $\widehat{\mathbb{U}}$ is the subspace of $\mathbb{U}$ and essentially a linear space formed by $n$ basis functions $\{\Psi_1, \cdots, \Psi_n\}$. Thus, the above variational formalization of PDE can be transformed into an approximated problem, namely, find $\widehat{u} \in \widehat{\mathbb{U}}$, s.t. $D(\widehat{u}, \widehat{v}) - B(\widehat{v}) = 0, \forall \widehat{v} \in \widehat{\mathbb{U}}$.

Usually, $\{\Psi_1, \cdots, \Psi_n\}$ are defined as the linear interpolation functions of a region:

$$\Psi_i(\mathbf{x}) = \begin{cases} 1 & \text{if } \mathbf{x} = \mathbf{x}_i, \\ \text{Linear-Interpolation}(\mathbf{x}) & \text{if } \mathbf{x} \in \text{Region}(\mathbf{x}_i) \\ 0 & \text{if } \mathbf{x} \in \Omega\backslash\text{Region}(\mathbf{x}_i) \end{cases}, \quad i = 1, \cdots, n. \quad (12)$$

$\text{Region}(\mathbf{x}_i)$ denotes the triangular mesh adjacent to $\mathbf{x}_i$ and $\Psi_i$ is zero on the boundary.

Since $\widehat{\mathbb{U}}$ is a linear space formed by $n$ basis functions $\{\Psi_1, \cdots, \Psi_n\}$, $\widehat{u} = \sum_{i=1}^n u_i\Psi_i$ and $\widehat{v} = \sum_{i=1}^n v_i\Psi_i$. Thus, the PDE is approximated by solving the following equation set:

$$\sum_{j=1}^n D(\Psi_j, \Psi_i)u_j - B(\Psi_i) = 0, \quad i = 1, \cdots, n. \quad (13)$$

It is worth noticing that according to the definition in Equation 12, basis function $\Psi_i$ is zero in all the other nodes $\mathbf{x}_j, i \neq j$. Thus, the $i$-th coefficient $u_i$ is also the approximated solution value on node $\mathbf{x}_i$.

According to the updating strategy of the Jacobi iterative method (Goldstine et al., 1959), we can directly obtain:

$$u_j^{(k+1)} = \frac{1}{D(\Psi_j, \Psi_j)}\left(b_j - \sum_{i\neq j} D(\Psi_i, \Psi_j)u_i^{(k)}\right), \quad (14)$$

where $b_j = B(\Psi_j)$ is a constant related to the external force $f$ and $u_j^{(k+1)}$ represents the coefficient value in the $j$-th step, which is also equal to the solution value in the $j$-th node. $\qquad\square$

## B.2 Gradient Correlation (Theorem 3.6)

According to Definition 3.4, if $\mathbf{x}$ and $\mathbf{x}'$ are adjacent and $D_{\text{PINN}}(\mathbf{x}, \mathbf{x}')$, defined in Equation 4, is less than a empirically defined threshold $\epsilon$, we consider that propagation failure has occurred between $\mathbf{x}$ and $\mathbf{x}'$. In this theorem, we want to prove that the propagation failure is equivalent to a small gradient correlation $G_{u_\theta}(\mathbf{x}, \mathbf{x}')$ between two adjacent points $\mathbf{x}$ and $\mathbf{x}'$.

*Proof.* We notice that the PINN $u_\theta$ can be regarded as a multivariate function with respect to the parameters $\theta$ and the input variables $\mathbf{x}$, i.e., $u(\theta, x) = u_\theta(x)$. Since $u(\theta, x)$ is infinitely differentiable with respect to both $\theta$ and $\mathbf{x}$, and let $\lambda$ be a sufficiently small step size, we proceed to rewrite the Equation 4 by employing a

Taylor expansion centered at $(\theta, \mathbf{x}')$,

$$
\begin{aligned}
D_{\text{PINN}}(\mathbf{x}, \mathbf{x}') &= \lim_{\lambda \to 0} \frac{\left\| u_\theta(\mathbf{x}') - u_{\theta - \lambda \frac{\partial u_\theta}{\partial \theta}\big|_{\mathbf{x}}}(\mathbf{x}') \right\|}{\lambda} \\
&= \lim_{\lambda \to 0} \frac{\left\| u(\theta, \mathbf{x}') - u(\theta - \lambda \frac{\partial u_\theta}{\partial \theta}(\mathbf{x}), \mathbf{x}') \right\|}{\lambda} \\
&= \lim_{\lambda \to 0} \frac{\left\| \left\langle \frac{\partial u_\theta}{\partial \theta}(\mathbf{x}'), \lambda \frac{\partial u_\theta}{\partial \theta}(\mathbf{x}) \right\rangle + \mathcal{O}(\lambda^2) \right\|}{\lambda} \\
&= \lim_{\lambda \to 0} \frac{\lambda G_{u_\theta}(\mathbf{x}, \mathbf{x}') + \mathcal{O}(\lambda^2)}{\lambda} \\
&= G_{u_\theta}(\mathbf{x}, \mathbf{x}').
\end{aligned}
\tag{15}
$$

Therefore, since the functions $D_{\text{PINN}}(\mathbf{x}, \mathbf{x}')$ and $G_{u_\theta}(\mathbf{x}, \mathbf{x}')$ is equivalent, the sufficient smallness of $D$ guarantees the sufficient smallness of $G$, and the reverse is also true. $\qquad\square$

It worth noticing that, although $D_{\text{PINN}}(\mathbf{x}, \mathbf{x}')$ and $G_{u_\theta}(\mathbf{x}, \mathbf{x}')$ are numerically equal, they are under different perspectives. Specifically, $D_{\text{PINN}}(\mathbf{x}, \mathbf{x}')$ is derived based on the physical meaningful of FEMs, which can perfectly reflect the definition of "stiffness matrix", namely the affect to position $\mathbf{x}'$ when $\mathbf{x}$ has a unit displacement or force. In contrast, the definition of $G_{u_\theta}(\mathbf{x}, \mathbf{x}')$ describes the property of PINN models, which provides a clearer understanding in the deep learning context.

### B.3    Gradient Correlation Improvement (Theorem 3.10)

This theorem demonstrates that uniting region gradients can boost the gradient correlation among nearby points. This theorem can be proved under the Assumption 3.9, which assumes the positive gradient correlation of nearby points.

*Proof.* To demonstrate the effectiveness of uniting region gradients, it is equivalent to proving that,

$$
\begin{aligned}
&G_{u_\theta}(\mathbf{x}, \mathbf{x}') \leq G_{u_\theta^{\text{region}}}(\mathbf{x}, \mathbf{x}') \\
\Leftrightarrow\ &G_{u_\theta}(\mathbf{x}, \mathbf{x}') \leq G_{u_\theta(\mathbf{x}) + \frac{1}{k}\sum_{i=1}^k u_\theta(\mathbf{x}+\boldsymbol{\delta}_i)}(\mathbf{x}, \mathbf{x}'),
\end{aligned}
\tag{16}
$$

which can be further expanded as follows

$$
\begin{aligned}
\Leftrightarrow\ &\left\| \left\langle \frac{\partial u_\theta}{\partial \theta}\Big|_{\mathbf{x}}, \frac{\partial u_\theta}{\partial \theta}\Big|_{\mathbf{x}'} \right\rangle \right\| \leq \left\| \left\langle (\frac{\partial u_\theta}{\partial \theta}(\mathbf{x}) + \frac{1}{k}\sum_{i=1}^k \frac{\partial u_\theta}{\partial \theta}(\mathbf{x}+\boldsymbol{\delta}_i)), (\frac{\partial u_\theta}{\partial \theta}(\mathbf{x}') + \frac{1}{k}\sum_{i=1}^k \frac{\partial u_\theta}{\partial \theta}(\mathbf{x}'+\boldsymbol{\delta}_i)) \right\rangle \right\| \\
\Leftrightarrow\ &\left\langle \frac{\partial u_\theta}{\partial \theta}(\mathbf{x}), \frac{\partial u_\theta}{\partial \theta}(\mathbf{x}') \right\rangle \leq \left\| \left\langle (\frac{\partial u_\theta}{\partial \theta}(\mathbf{x}) + \frac{1}{k}\sum_{i=1}^k \frac{\partial u_\theta}{\partial \theta}(\mathbf{x}+\boldsymbol{\delta}_i)), (\frac{\partial u_\theta}{\partial \theta}(\mathbf{x}') + \frac{1}{k}\sum_{i=1}^k \frac{\partial u_\theta}{\partial \theta}(\mathbf{x}'+\boldsymbol{\delta}_i)) \right\rangle \right\|.
\end{aligned}
\tag{17}
$$

Here the third $\Leftrightarrow$ is due to the condition that $\|\mathbf{x} - \mathbf{x}'\| \leq \frac{R}{3}$, thus $\left\langle \frac{\partial u_\theta}{\partial \theta}(\mathbf{x}), \frac{\partial u_\theta}{\partial \theta}(\mathbf{x}') \right\rangle \geq 0$.

From the pre-defined perturbations $\{\boldsymbol{\delta}_i\}_{i=1}^k$ with $\|\boldsymbol{\delta}_i\| \leq \frac{R}{3}$ and the given $\mathbf{x}$ and $\mathbf{x}'$ such that $\|\mathbf{x} - \mathbf{x}'\| \leq \frac{R}{3}$, it follows that,

$$
\|(\mathbf{x} + \delta_i) - \mathbf{x}'\| \leq R,\ \|\mathbf{x} - (\mathbf{x}' + \delta_j)\| \leq R,\ \|(\mathbf{x} + \delta_i) - (\mathbf{x}' + \delta_j)\| \leq R,\ \forall i, j \in \{1, \ldots, k\}.
\tag{18}
$$

Given the choice of $R$ in Assumption 3.9, we obtain that

$$
\begin{aligned}
\Bigg\| \Bigg\langle \Big( &\frac{\partial u_\theta}{\partial \theta}(\mathbf{x}) + \frac{1}{k}\sum_{i=1}^{k}\frac{\partial u_\theta}{\partial \theta}(\mathbf{x}+\boldsymbol{\delta}_i)\Big), \Big(\frac{\partial u_\theta}{\partial \theta}(\mathbf{x}') + \frac{1}{k}\sum_{i=1}^{k}\frac{\partial u_\theta}{\partial \theta}(\mathbf{x}'+\boldsymbol{\delta}_i)\Big) \Bigg\rangle \Bigg\| \\
= & \left\langle \frac{\partial u_\theta}{\partial \theta}(\mathbf{x}), \frac{\partial u_\theta}{\partial \theta}(\mathbf{x}') \right\rangle + \left\langle \frac{\partial u_\theta}{\partial \theta}(\mathbf{x}), \frac{1}{k}\sum_{i=1}^{k}\frac{\partial u_\theta}{\partial \theta}(\mathbf{x}'+\boldsymbol{\delta}_i) \right\rangle \\
& + \left\langle \frac{1}{k}\sum_{i=1}^{k}\frac{\partial u_\theta}{\partial \theta}(\mathbf{x}+\boldsymbol{\delta}_i), \frac{\partial u_\theta}{\partial \theta}(\mathbf{x}') \right\rangle + \left\langle \frac{1}{k}\sum_{i=1}^{k}\frac{\partial u_\theta}{\partial \theta}(\mathbf{x}+\boldsymbol{\delta}_i), \frac{1}{k}\sum_{i=1}^{k}\frac{\partial u_\theta}{\partial \theta}(\mathbf{x}'+\boldsymbol{\delta}_i) \right\rangle \\
\geq & \left\langle \frac{\partial u_\theta}{\partial \theta}(\mathbf{x}), \frac{\partial u_\theta}{\partial \theta}(\mathbf{x}') \right\rangle .
\end{aligned}
\tag{19}
$$

Consequently, the conclusion in Equation 16 and Theorem 3.10 has been proven. $\qquad\square$

### B.4 Justification of Assumption 3.9

**Theoretical understanding**  Firstly, we want to highlight that if we change the positive constraint of region size $R$ in the assumption to "non-negative", Assumption 3.9 is always true. This can be directly proved by the following derivation:

$$
\left\langle \frac{\partial u_\theta}{\partial \theta}\Big|_{\mathbf{x}}, \frac{\partial u_\theta}{\partial \theta}\Big|_{\mathbf{x}} \right\rangle \geq 0.
\tag{20}
$$

Further, we still consider the positive constraint of region size $R$. If $\|\frac{\partial u_\theta}{\partial \theta}\big|_{\mathbf{x}}\| \neq 0$ at $\mathbf{x}$, then $\left\langle \frac{\partial u_\theta}{\partial \theta}\big|_{\mathbf{x}}, \frac{\partial u_\theta}{\partial \theta}\big|_{\mathbf{x}} \right\rangle >$

0. According to the boundedness of $\frac{\partial^2 u_\theta}{\partial \theta \partial \mathbf{x}}$, there must exist a region $R_{\mathbf{x}} = \dfrac{\left\langle \frac{\partial u_\theta}{\partial \theta}\big|_{\mathbf{x}}, \frac{\partial u_\theta}{\partial \theta}\big|_{\mathbf{x}} \right\rangle}{2\left\langle \frac{\partial u_\theta}{\partial \theta}\big|_{\mathbf{x}}, \frac{\partial^2 u_\theta}{\partial \theta \partial \mathbf{x}} \right\rangle} > 0$ s.t. $\forall \mathbf{x}' \in \Omega$, if

$\|\mathbf{x}' - \mathbf{x}\| \leq R_{\mathbf{x}}$, we have

$$
\begin{aligned}
\left\langle \frac{\partial u_\theta}{\partial \theta}\Big|_{\mathbf{x}}, \frac{\partial u_\theta}{\partial \theta}\Big|_{\mathbf{x}'} \right\rangle &= \left\langle \frac{\partial u_\theta}{\partial \theta}\Big|_{\mathbf{x}}, \frac{\partial u_\theta}{\partial \theta}\Big|_{\mathbf{x}} + (\mathbf{x}' - \mathbf{x})\frac{\partial^2 u_\theta}{\partial \theta \partial \mathbf{x}} + \mathcal{O}((\mathbf{x}' - \mathbf{x})^2) \right\rangle \\
&\geq \left\langle \frac{\partial u_\theta}{\partial \theta}\Big|_{\mathbf{x}}, \frac{\partial u_\theta}{\partial \theta}\Big|_{\mathbf{x}} \right\rangle - R_{\mathbf{x}}\left\langle \frac{\partial u_\theta}{\partial \theta}\Big|_{\mathbf{x}}, \frac{\partial^2 u_\theta}{\partial \theta \partial \mathbf{x}} \right\rangle \\
&= \frac{1}{2}\left\langle \frac{\partial u_\theta}{\partial \theta}\Big|_{\mathbf{x}}, \frac{\partial u_\theta}{\partial \theta}\Big|_{\mathbf{x}} \right\rangle \\
&> 0.
\end{aligned}
\tag{21}
$$

Thus, the unguaranteed part in Assumption 3.9 is $R = \min_{\mathbf{x}\in\Omega} R_{\mathbf{x}} > 0$, namely *is there a unified region size for all collocation points*. The following experiment statistics can well verify this question.

**Experimental statistics**  We also count the proportion of points that satisfy Assumption 3.9 in each PDE. Specifically, we take $10^4$ equally spaced points at each PDE. For each collocation point, we consider its gradient correlation with nearby points, whose distance is around $10^{-2}$. As presented in Table 4, if we set $R$ as $10^{-2}$, we can find all the collocation points are under positive region gradient correlation, indicating that Assumption 3.9 can be well guaranteed in practice.

Table 4: Statistics for Assumption 3.9.

| Statistics of $10^4$ Points | Convection | 1D-Reaction | Allen-Cahn | 1D-Wave |
|---|---|---|---|---|
| Positive Ratio of region Gradient Correlations | 100% | 100% | 100% | 100% |

# C   Implementation Details

In this section, we will provide details about benchmarks, experiment settings, model configurations and evaluation metrics.

## C.1   Benchmark Description

As shown in Figure 11, we evaluate ProPINN on six tasks, which include four standard benchmarks: Convection, 1D-Reaction, Allen-Cahn and 1D-Wave and two complex physics modeling tasks: Karman Vortex and Fluid Dynamics. Here are the details of each benchmark.

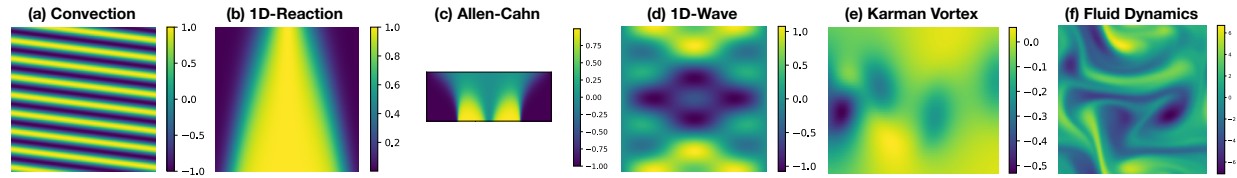

Figure 11: Summary of benchmarks. We visualize the solution map for each solving task.

**Convection**   This problem describes a hyperbolic PDE, which can be formalized as follows:

$$\text{Equation constraint: } \frac{\partial u}{\partial t} + \beta \frac{\partial u}{\partial x} = 0, \ x \in (0, 2\pi), t \in (0, 1),$$
$$\text{Initial condition: } u(x, 0) = \sin(x), \ x \in [0, 2\pi], \tag{22}$$
$$\text{Boundary condition: } u(0, t) = u(2\pi, t), \ t \in [0, 1],$$

where the convection coefficient $\beta$ is set as 50. Its analytic solution is $u(x, t) = \sin(x - \beta t)$. As presented in Figure 11(a), this problem involves rapid variations along the temporal dimension, making it hard for neural networks to approximate. Thus, this problem is widely tested in characterizing PINN failure modes (Krishnapriyan et al., 2021) and evaluating new architectures (Zhao et al., 2024; Nagda et al., 2024).

**1D-Reaction**   This problem is a non-linear PDE for chemical reactions, whose equation is formalized as:

$$\text{Equation constraint: } \frac{\partial u}{\partial t} - \rho u(1 - u) = 0, \ x \in (0, 2\pi), t \in (0, 1),$$
$$\text{Initial condition: } u(x, 0) = \exp\left(-\frac{(x - \pi)^2}{2(\pi/4)^2}\right), \ x \in [0, 2\pi], \tag{23}$$
$$\text{Boundary condition: } u(0, t) = u(2\pi, t), \ t \in [0, 1],$$

where the PDE coefficient $\rho$ is set as 5. The analytic solution for this PDE is $u(x, t) = \frac{h(x)e^{\rho t}}{h(x)e^{\rho t}+1-h(x)}$, where $h(x) = \exp\left(-\frac{(x-\pi)}{2(\pi/4)^2}\right)$. As shown in Figure 11(b), this task presents rapid variations in some areas, making it hard to solve (Krishnapriyan et al., 2021). We experimented with this problem following PINNsFormer (Zhao et al., 2024).

**Allen-Cahn**   This problem is a typical reaction-diffusion equation, which is defined as:

$$\text{Equation constraint: } \frac{\partial u}{\partial t} - 0.0001\frac{\partial^2 u}{\partial x^2} + 5u^3 - 5u = 0, \ x \in (-1, 1), t \in (0, 1),$$
$$\text{Initial condition: } u(x, 0) = x^2 \cos(\pi x), \ x \in [-1, 1],$$
$$\text{Boundary condition: } u(-1, t) = u(1, t), \ t \in [0, 1], \tag{24}$$
$$\text{Boundary condition: } \frac{\partial u(-1, t)}{\partial x} = \frac{\partial u(1, t)}{\partial x}, \ t \in [0, 1].$$

Since this PDE does not have an analytic solution, following previous studies (Raissi et al., 2019), we adopt the results pre-calculated by traditional spectral methods (Kopriva, 2009) as the reference. As illustrated

in Figure 11(c), this task also includes the sharp area, making it usually studied as PINN failure modes (Krishnapriyan et al., 2021).

**1D-Wave**   This problem is a hyperbolic PDE, which involves high-order derivatives in its equation constraint. The exact governing equation is defined as follows:

$$
\begin{aligned}
\text{Equation constraint: } & \frac{\partial^2 u}{\partial t^2} - 4\frac{\partial^2 u}{\partial x^2} = 0, \ x \in (0,1), t \in (0,1), \\
\text{Initial condition: } & u(x,0) = \sin(\pi x) + \frac{1}{2}\sin(\beta\pi x), \ x \in [0,1], \\
\text{Initial condition: } & \frac{\partial u(x,0)}{\partial t} = 0, \ x \in [0,1], \\
\text{Boundary condition: } & u(0,t) = u(1,t) = 0, \ t \in [0,1],
\end{aligned}
\tag{25}
$$

where its periodic coefficient $\beta$ is set as 3 and the analytic solution is $u(x,t) = \sin(\pi x)\cos(2\pi t) + \frac{1}{2}\sin(\beta\pi x)\cos(2\beta\pi t)$. This solution presents periodic patterns, which require the neural network to fit a periodic output space (Figure 11(d)).

**Karman Vortex**   This task describes the incompressible fluid moving past a cylinder, which involves the famous Karman vortex street phenomenon (Wille, 1960) as shown in Figure 11(e) and is governed by the following Navier-Stokes equations:

$$
\begin{aligned}
\frac{\partial u}{\partial t} + (u\frac{\partial u}{\partial x} + v\frac{\partial u}{\partial y}) &= -\frac{\partial p}{\partial x} + 0.01(\frac{\partial^2 u}{\partial x^2} + \frac{\partial^2 u}{\partial y^2}) \\
\frac{\partial v}{\partial t} + (u\frac{\partial v}{\partial x} + v\frac{\partial v}{\partial y}) &= -\frac{\partial p}{\partial y} + 0.01(\frac{\partial^2 v}{\partial x^2} + \frac{\partial^2 v}{\partial y^2}) \\
\frac{\partial u}{\partial x} + \frac{\partial v}{\partial y} &= 0,
\end{aligned}
\tag{26}
$$

where $u, v$ represent the velocity along the x-axis and y-axis. $p$ denotes the pressure field. Since we cannot obtain the analytic solution of the Navier-Stokes equations, we experiment with the high-resolution data calculated by the spectral/hp-element solver NekTar (Karniadakis & Sherwin, 2005) following Raissi et al..

Specifically, the generated fluid sequence contains 200 frames. The task is to reconstruct the pressure field $p$ with physics loss defined in the above, which contains the above three equations and the supervision of ground truth velocity.

**Fluid Dynamics**   This problem is from the well-established PINN framework JAX-PI (Wang et al., 2023b), using the 2D incompressible Navier-Stokes equations in fluid dynamics and appropriate initial conditions to simulate fluid flow in a torus. The governed PDEs and initial conditions we used are shown as:

$$
\begin{aligned}
\frac{\partial w}{\partial t} + (u\frac{\partial w}{\partial x} + v\frac{\partial w}{\partial y}) &= \frac{1}{\text{Re}}(\frac{\partial^2 w}{\partial x^2} + \frac{\partial^2 w}{\partial y^2}), && (t,x,y) \in [0,T] \times \Omega \\
\frac{\partial u}{\partial x} + \frac{\partial v}{\partial y} &= 0, && (t,x,y) \in [0,T] \times \Omega \\
w(0,x,y) &= w_0(x,y), \ (x,y) \in \Omega
\end{aligned}
\tag{27}
$$

where $\mathbf{u} = (u,v)$ is the two-dimensional velocity vector of the fluid, $w = \nabla \times \mathbf{u}$ is the fluid vorticity, with $T$ set to 10s, $\Omega$ set to $[0,2\pi]^2$, and the Reynolds number Re set to 100. $w_0$ denotes the given initial condition of vorticity.

The task is to simulate the future 10 seconds of fluid vorticity field $w$ solely based on the initial condition. However, the convective term $(\mathbf{u} \cdot \nabla)\mathbf{u}$ in the Navier-Stokes equations is nonlinear, which can lead to chaotic behavior. Thus, small changes in initial conditions can result in significantly different outcomes, making this task extremely difficult (Constantin & Foiaş, 1988). As presented in Figure 11(f), the physics field in this task is quite complex.

## C.2 Experiment Settings

We repeat all the experiments three times and report the average performance in the main text. All the experiments are conducted on a single A100 40GB GPU. Here are detailed configurations for each benchmark. The standard deviations can be found in Appendix G.

**Standard benchmarks**   For Convection, 1D-Reaction and 1D-Wave, we following the experiment settings in PINNsFormer (Zhao et al., 2024). Specifically, each experiment selects $101 \times 101$ collocation points in the input domain and sets loss weights $\lambda_* = 1$. All the models are trained with L-BFGS optimizer (Liu & Nocedal, 1989) for 1,000 iterations in PyTorch (Paszke et al., 2019). As for Allen-Cahn, we also implement this PDE in PyTorch with all the configurations same as PINNsFormer but set $\lambda_{\mathrm{res}} = \lambda_{\mathrm{bc}} = 1$ and $\lambda_{\mathrm{ic}} = 10$ for all models to fit the complex initial condition.

**Karman Vortex**   As we stated before, this task is supervised by the equation constraints and ground truth of velocity. Concretely, we randomly select 2500 collocation points from the whole spatiotemporal sequence and train the model with L-BFGS optimizer for 1,000 iterations in PyTorch with loss weights $\lambda_* = 1$ following PINNsFormer (Zhao et al., 2024).

**Fluid Dynamics**   In this task, we use the JAX-PI framework[1]. Here are the detailed settings. Firstly, to tackle the long temporal interval, we split the temporal domain into 10 windows and each window is trained separately in sequence. For each window, a total of 150,000 training steps are used, with each training step sampling uniformly distributed points in both the temporal and spatial domains, totaling 4096 coordinates $(t, x, y)$. Then, the corresponding governed PDEs and initial conditions (Equation 27) are used as loss functions for training. Specifically, for the first window, the initial condition is the value of the exact solution at the beginning time, and subsequent windows use the solution at the last timestamp from the previous window as the initial condition. Although this time-marching strategy may lead to error accumulation due to the setting of initial conditions, excellent models and training methods can maintain very low relative errors even in the last window. Note that JAX-PI also utilizes some tricks to ensure the final performance, such as random Fourier feature embedding (Tancik et al., 2020). We also maintain these tricks in all the models to ensure that the only variable is model architecture for rigorous comparison. In this task, we use Adam (Kingma & Ba, 2015) with a learning rate of 0.001 for optimization.

## C.3 Model Configuration

**ProPINN**   As highlighted in the main text, we insist on the lightweight design in ProPINN. Specifically, the projection layer $\mathcal{P}$ involves two linear layers with an in-between activation function, where the input dimension $(d + 1)$ is firstly projected to 8 and then to 32, which is the same as the embedding size of PINNsFormer (Zhao et al., 2024). And the multi-region mixing layer $\mathcal{M}$ also involves two linear layers with an in-between activation, which is only applied to the region dimension. Specifically, $\mathcal{M}$ will first project the #scale scales to 8 and then to 1 after an activation layer. As for the final projection layer $\mathcal{H}$, it consists of three linear layers with inner activations, whose hidden dimension is set as 64.

**Baselines**   In our experiments, we also compare ProPINN with seven baselines. Here are our implementation details for these baselines:

- For vanilla PINN (Raissi et al., 2019), QRes (Bu & Karpatne, 2021), FLS (Wong et al., 2022) and PINNsFormer (Zhao et al., 2024), we follow the PyTorch implementation of these models provided in PINNsFormer and reimplement them in JAX to fit the Fluid Dynamics task in JAX-PI (Wang et al., 2023b). Specifically, vanilla PINN, QRes and FLS are all with 4 layers with 512 hidden channels. PINNsFormer contains 1 encoder layer and 1 decoder layer with 32 hidden channels for the attention mechanism and 512 hidden channels for the feedforward layer.

- As for SetPINN (Nagda et al., 2024), we implement this model based on their official paper, which can reproduce the results reported in their paper. Also, we reimplement it in JAX for the Fluid Dynamics

---

[1] https://github.com/PredictiveIntelligenceLab/jaxpi

task. Specifically, same to the official configuration in PINNsFormer, SetPINN is experimented with 1 encoder layer and 1 decoder layer, which contains 32 hidden channels for the attention mechanism and 512 hidden channels for the feedforward layer.

- For KAN (Liu et al., 2024), we adopt their official code and set hyperparameters following (Wu et al., 2024).

- For PirateNet (Wang et al., 2024a), its official implementation is in JAX. Thus, we directly test their official version in the Fluid Dynamics task and reimplement it in PyTorch for other benchmarks. Specifically, it contains 256 hidden channels for representations and 4 layers following its official configuration.

**Reproduction of baselines**  For all baselines, we made a great effort to reproduce their performance. Especially, the above-mentioned hyperparameters can perfectly reproduce their official results. Thus, in the shared tasks, we directly report the official results of baselines. As for new tasks (e.g. Allen-Cahn, Fluid Dynamics) without official performance of baselines, we use the same configuration as the previous reproduction. Notably, ProPINN adopts the same model configuration for all PDEs. Thus, the comparison is fair and rigorous.

### C.4  Metrics

As we stated before, we evaluate the model-predicted solution based on relative L1 error (rMAE) and relative Root Mean Square Error (rRMSE). For ground truth $u$ and model prediction $u_\theta$, these two metrics can be calculated as follows:

$$\text{rMAE: } \sqrt{\frac{\sum_{i=1}^{n} |u_\theta(\mathbf{x}_i) - u(\mathbf{x}_i)|}{\sum_{i=1}^{n} |u(\mathbf{x}_i)|}} \quad \text{rRMSE: } \sqrt{\frac{\sum_{i=1}^{n} (u_\theta(\mathbf{x}_i) - u(\mathbf{x}_i))^2}{\sum_{i=1}^{n} (u(\mathbf{x}_i))^2}}, \tag{28}$$

where $\{\mathbf{x}_i\}_{i=1}^{n}$ are selected collocation points for evaluation.

## D  Hyperparameter Analysis

As a supplement to ablations in Figure 7 of the main text, we also test the model performance under different hyperparameter configurations, including the number of perturbations at each scale $(k_1, k_2, k_3)$, size of perturbation region $(R_1, R_2, R_3)$ and the number of scales #scale. The results are presented in Figure 12, where we can obtain the following observations.

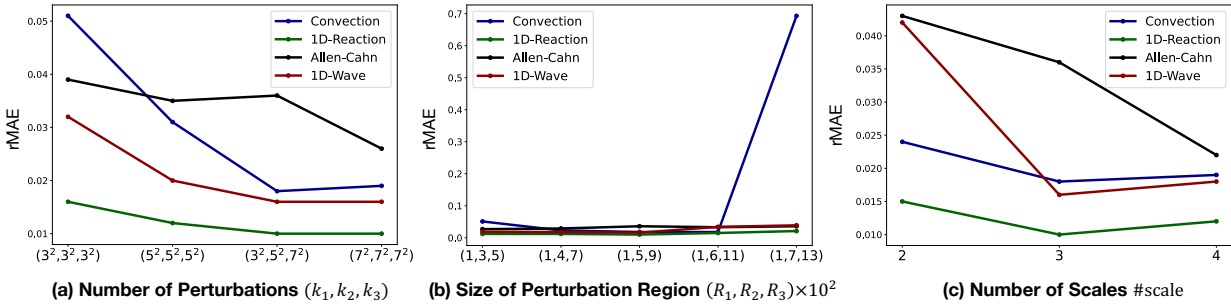

**(a) Number of Perturbations** $(k_1, k_2, k_3)$   **(b) Size of Perturbation Region** $(R_1, R_2, R_3) \times 10^2$   **(c) Number of Scales** #scale

Figure 12: Model analysis with respect to different hyperparameter configurations of ProPINN. For (c), two scales correspond to $(R_1, R_2) = (0.01, 0.05)$ and four scales correspond to $(R_1, R_2, R_3, R_4) = (0.01, 0.03, 0.07, 0.09)$.

*(i) More perturbations will generally boost the model performance.* As stated in Section 3.2, we adopt differential perturbation to unite gradients of region points. Thus, adding perturbation points will enhance the connection of region gradients, thereby benefiting the final performance, while this will also bring more

computation costs. Thus, we choose $(k_1, k_2, k_3) = (3^2, 5^2, 7^2)$ for a better trade-off between efficiency and performance.

*(ii) The size of the perturbation region is up to the PDE property.* As Remark 3.5 discussed, the gradient correlations only consider the points within a region. Therefore, in the Convection equation that involves the rapid variation in the PDE solution, enlarging the perturbation size to $R_3 = 0.13$ (the whole domain is $[0, 2\pi] \times [0, 1]$) may introduce noise to the learning process, making the final performance degenerate seriously. Also, it is observed that ProPINN performs relatively steadily on all other benchmarks in different perturbation regions. Thus, we believe the effect of this hyperparameter is up to a certain PDE. And our design in choosing $(R_1, R_2, R_3)$ as $(0.01, 0.05, 0.09)$ can be a generalizable choice.

*(iii) Adding scales can also boost the model performance.* As shown in Figure 7 of the main text, only one scale in ProPINN will bring a serious performance drop. Further, in Figure 12(c), we increase the number of scales from 2 to 4. It is observed that the model performance is generally improved, especially for Allen-Cahn, which has a complex solution, and thereby can benefit more from adding scales. These results further demonstrate the effectiveness of our design in multi-region mixing.

## E    More Ablations

To supplement Figure 7, we provide more ablations on model performance and design here.

**Performance under aligned efficiency**    As shown in Figure 5, some single-point-processing architectures, such as vanilla PINN, may present very significant speed. To ensure a more comprehensive comparison, we also include the performance comparison under the aligned efficiency, where we increase the number of layers and the training iterations. From Table 5, we can find that adding layers cannot improve the final performance. Actually, as presented in Figure 7(d) of the main text, PINN fails when scaling up the model size. That is why a larger model will bring worse performance in the following experiments. About training iterations, as discussed in this paper, PINN suffers from propagation failure, which means some areas cannot receive the correct supervision; that is why increasing the training iterations does not bring benefits to PINN.

Table 5: Model performance under aligned efficiency. To ensure a comprehensive comparison, we further test Vanilla PINN under more training iterations and more layers.

| | Training Iterations | | | Model Efficiency | |
|---|---|---|---|---|---|
| Convection rMAE | 1000 iters | 2000 iters | 4000 iters | GPU Memory (GB) | s / 1000iters |
| Vanilla PINN | 0.778 | 0.778 | 0.778 | 1.06 | 18.62 |
| Vanilla PINN + 2 Layers | 0.825 | 0.825 | 0.825 | 2.00 | 39.09 |
| ProPINN | 0.018 | 0.008 | 0.008 | 1.72 | 37.76 |

**Adopt other types of optimizers**    In this paper, we strictly follow the well-established benchmarks and L-BFGS for standard benchmarks and Karman Vortex, Adam for Fluid Dynamics, which is the same as the previous research (Nagda et al., 2024) and JAX-PI. All the baselines and ProPINN are under the same training strategy to ensure a fair comparison. Actually, there are other choices of optimizers, such as the combination of Adam and L-BFGS. To ensure a comprehensive comparison, we also compare different models under the Adam+L-BFGS setting. As shown below, under this new choice of Adam+L-BFGS, ProPINN is still the best model across all four benchmarks. Besides, comparing Table 6 and Table 2, we can find that pure L-BFGS can already achieve a good performance for all models. Since this paper only focuses on the model architecture rather than PINN optimizers, we would like to leave more discussion about optimizers as our future work.

**Adopt gradient correlation as regularization term**    As stated in Theorem 3.6, we prove that the root cause of propagation failure is a lower gradient correlation. Thus, it is a trivial solution to add gradient correlation as a regularization term for the PINN loss. Note that gradient correlation needs to calculate the gradients of model parameters on different collocation points $\frac{\partial u_\theta}{\partial \theta}(x)$. Firstly, it will take around 60s on a single A100 GPU to calculate gradient correlations of 10000 points of Convection in each step. Secondly,

Table 6: Model comparison under the Adam+L-BFGS setting, where we first train the model with Adam for 100 iterations and then optimize the model with L-BFGS for 1000 iterations.

| Model rRMSE under Adam+L-BFGS | Convection | 1D-Reaction | Allen-Cahn | 1D-Wave |
|---|---|---|---|---|
| Vanilla PINN (Raissi et al., 2019) | 0.823 | 0.982 | 0.576 | 0.381 |
| QRes (Bu & Karpatne, 2021) | 0.852 | 0.573 | 0.957 | 0.178 |
| FLS (Wong et al., 2022) | 0.693 | 0.049 | 0.651 | 0.186 |
| KAN (Liu et al., 2024) | 0.873 | 0.057 | 0.581 | 0.221 |
| PirateNet (Wang et al., 2024a) | 1.294 | 0.046 | 0.136 | 0.511 |
| PINNsFormer (Zhao et al., 2024) | 0.032 | 0.027 | 0.532 | 0.489 |
| SetPINN (Nagda et al., 2024) | 0.031 | 0.046 | 0.597 | 0.332 |
| **ProPINN (Ours)** | **0.020** | **0.020** | **0.087** | **0.016** |
| Promotion | 35% | 26% | 36% | 91% |

the optimization of gradient correlation loss will require calculating the second-order derivative of model parameters. Thus, we do not think this is a practical design.

Table 7: Comparison between ProPINN and gradient-correlation regularization term on Convection. "+GCL" means with direct gradient correlation loss.

| Model | rMAE | GPU Memory (GB) | Running Time (s/1000iters) |
|---|---|---|---|
| Vanilla PINN (2019) | 0.778 | 1.06 | 18.62 |
| Vanilla PINN (2019) + GCL | 0.528 | 2.31 | 65.17 |
| **ProPINN (Ours)** | **0.018** | **1.72** | **37.76** |

Here, we do some simplification to enable training of adding gradient correlation as a regularization term. Specifically, we split the collocation points into two interlaced sets and add gradient correlation between these two sets as a regularization term with a weight of 0.001. As shown in Table 7, this design is slightly better than PINN, but it is more time-consuming and worse than ProPINN.

**Other architecture choices to improve gradient correlation**   PINNsFormer (Nagda et al., 2024) and SetPINN (Nagda et al., 2024) explicitly capture the correlation among different collocation points. As shown in Table 2 and Figure 5, these two models cannot beat ProPINN in performance and efficiency. This may be because these two models only consider the "representation correlation", not the "gradient correlation".

Actually, we cannot figure out new architectures different from ProPINN that can guarantee gradient correlation (Theorem 3.10) and balance performance and efficiency. Thus, we believe that ProPINN is a favorable and theoretically guaranteed choice to enhance gradient correlation among nearby points.

**Quantitative boundary-localization analysis.**   To better evaluate the diagnostic value of gradient correlation, we adopt a boundary-localization perspective instead of directly comparing the correlation between diagnostic scores and the full error map point-wise. Specifically, we extract the high-error region from the error map, treat its boundary as the observable structure of propagation failure, and measure how closely the high-score regions of different diagnostics align with this boundary. This design is motivated by the fact that gradient correlation is a neighborhood-wise propagation quantity: it reflects whether supervision can propagate across nearby regions, and is therefore more naturally interpreted as an indicator of propagation barriers than as a predictor of point-wise error magnitude.

Under this analysis, low-gradient-correlation is, on average, closer to the error-derived high-error boundary than residual loss. This supports our claim that gradient correlation is more suitable for identifying failure boundaries or barrier locations, while residual loss mainly reflects local PDE violation.

Table 8: Average boundary-localization performance of residual loss and low-gradient-correlation. We define the boundary of the top-10% high-error region in the error map as the ground-truth high-error boundary, and treat the top-10% regions of the two diagnostic scores as the predicted boundary-prone regions. The table reports the average Distance-to-boundary over the four standard benchmarks. Lower values indicate that the diagnostic more accurately localizes the propagation barrier or failure boundary.

| Model | Residual Loss Distance | Low-Gradient Correlation Distance | Promotion |
|---|---|---|---|
| Vanilla PINN | 0.5047 | **0.3702** | 26.64% |
| ProPINN | 0.3413 | **0.2600** | 23.83% |

## F   More Showcases

In the main text, we have already provided the visualization comparison on Convection and 1D-Wave in Figure 4 and Fluid Dynamics in Figure 6. Here we also provide comparisons of the other three benchmarks in Figure 13 and Figure 14. It is easy to observe that ProPINN is better than PINNsFormer in handling the areas with rapid variations, including the corner of 1D-Reaction, the middle region of Allen-Cahn and the vortex of Karman Vortex, which come from its better propagation property. Besides, we can find that PirateNet under the single-point-processing paradigm fails to capture the vortex dynamics, indicating the inherent deficiency of single-point-processing models in complex physics simulations.

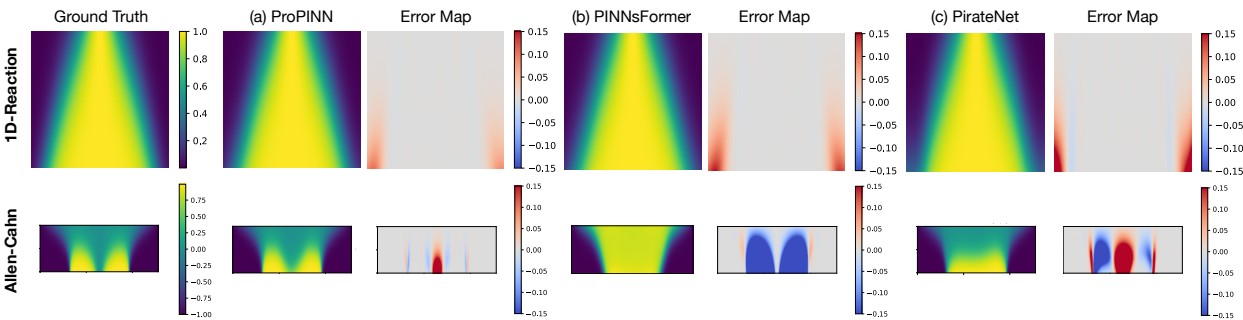

Figure 13: Visualizations of model approximated solution on the 1D-Reaction and Allen-Cahn benchmarks. Error map $(u_\theta - u)$ is also plotted.

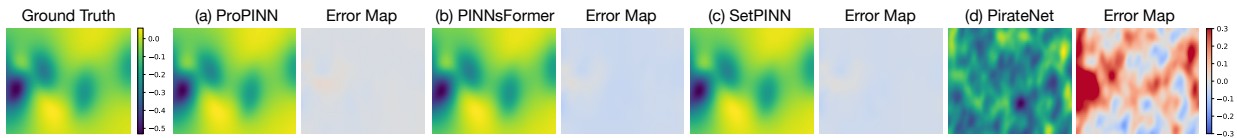

Figure 14: Comparison of model approximated solutions on the Karman Vortex benchmark. Error map $(u_\theta - u)$ is also plotted.

## G   Standard Deviations

All the experiments are repeated three times and the standard deviations are listed in Table 9. In all benchmarks, ProPINN surpasses the second-best model with high confidence.

## H   Limitations and Future Work

One potential limitation of ProPINN lies in its application to extremely high-dimensional PDEs, such as the Hamilton-Jacobi-Bellman equation in optimal control (Bardi et al., 1997) and the Schrodinger equation in quantum physics (Berezin & Shubin, 2012), which may involve millions of dimensions. These high-dimensional

Table 9: Standard deviations of ProPINN. We also list the performance of the second-best model. P-value $<$ 0.05 indicates that ProPINN surpasses the second-best model with high confidence.

| rMAE | Standard Benchmarks | | | | Complex Physics | |
|---|---|---|---|---|---|---|
| | Convection | 1D-Reaction | Allen-Cahn | 1D-Wave | Karman Vortex | Fluid Dynamics |
| Second-best | $0.023_{\pm0.002}$ (PINNsFormer) | $0.015_{\pm0.002}$ (PINNsFormer) | $0.098_{\pm0.007}$ (PirateNet) | $0.051_{\pm0.008}$ (PirateNet) | $0.287_{\pm0.08}$ (SetPINN) | $0.2362_{\pm0.013}$ (FLS) |
| ProPINN | $0.018_{\pm0.001}$ | $0.010_{\pm0.001}$ | $0.036_{\pm0.006}$ | $0.016_{\pm0.004}$ | $0.161_{\pm0.03}$ | $0.1834_{\pm0.010}$ |
| P-Value | 0.015 | 0.015 | 0.000 | 0.001 | 0.026 | 0.003 |

PDE-solving tasks will require the model to augment many points within each region, which may bring a huge computation overload for multi-region mixing.

This limitation can be resolved by integrating the "amortization" technique (Hu et al., 2023; Shi et al., 2024; Hu et al., 2024) with ProPINN, which can "amortize the computation over the optimization process via randomization". Since this paper mainly focuses on propagation failures of PINNs instead of high-dimensional issues and we have conducted extensive experiments in PINN failure modes and complex physics to verify the model effectiveness, we would like to leave the topic as our future work. Also, we would like to highlight that the favorable results in solving Navier-Stokes equations are already valuable for extensive real-world applications, such as aerodynamic simulation (McCormick, 1994).

## I  Broader Impacts

This paper provides an in-depth study of the propagation failure in PINNs, which is a crucial phenomenon of PINN optimization. We formally prove that the root cause of propagation failure is the lower gradient correlation, which can be inspiring for future research. Also, we present ProPINN as a practical PINN architecture with favorable efficiency and significant promotion over the previous methods. More importantly, ProPINN performs well in solving complex Navier-Stokes equations, which can be valuable for real-world applications. Since we purely focus on the research problem of PINNs, there are no potential negative social impacts or ethical risks.

