# OpenReview forum: "ProPINN: Demystifying Propagation Failures in Physics Informed Neural Networks"
_TMLR — Decision pending for TMLR_

### Review · Reviewer_6buy · 2026-04-07

**Summary Of Contributions:**

The paper studies propagation failure in PINNs, which is the inability of correct boundary and initial supervision to reach interior points. By analogy with FEM stiffness matrices, the authors define a PINN-specific stiffness co-efficient and prove that it equals the gradient correlation (Jacobian inner product) between nearby points, establishing low gradient correlation as the root cause. The authors then propose ProPINN, which uses differential perturbations in multi-scale regions with lightweight MLP mixing to share gradient paths among nearby points. Empirical results are promising: Experiments on six PDE benchmarks show around 46% average error reduction over Transformer-based PINNs at 2-3× lower cost.

Strength

`+` **This paper is well-motivated:** The FEM-to-PINN analogy is well constructed and gives us some actionable insight (gradient correlation as the root cause of propagation failure); this bridges numerical methods and deep learning perspectives.

`+` **The proposed method (ProPINN) is simple and effective.** We can see that it achieves large gains across all six PDE benchmarks while being 2-3× faster than Transformer-based alternatives. The ablation shows that non-differential perturbation hurts performance, which is quite convincing that gradient uniting (instead of representation augmentation) is the key mechanism that works.

`+` **Evaluation is pretty solid and comprehensive:** six diverse PDEs, seven baselines, ablations, scalability analysis, hyper-parameter sweeps, efficiency comparisons, and statistical significance tests with p-values. I am quite impressed.

Weakness: overall this paper is well-developed and I have a few comments on evaluation / certain theorem's framing

`-` On the hardest benchmark (Fluid Dynamics), PINNsFormer and SetPINN produce NaN and KAN is untested, so ProPINN is only compared against single-point-processing methods. The authors did not provide investigation on why Transformer baselines fail. I feel that the comparison is a bit incomplete there.

`-` The claim that gradient correlation is a "more precise criterion" than residual loss (page 2, paragraph 2 & the first bullet point) is only supported by visual inspection of Figure 1. A quantitative comparison (like spatial correlation with actual error maps) is needed to substantiate this central claim.

**Audience:**

Yes

**Audience Explanation:**

PINN training failures is a widely recognized problem and this paper offers both a principled diagnosis (gradient correlation as the root cause of propagation failure) and a practical + efficient architectural fix (ProPINN). The FEM comparison provides useful intuition that bridges the numerical analysis community and the deep learning community. The topic is timely given growing interest in scientific ML and the proposed method is simple + effective enough to be adopted by practitioners. Overall, this paper would interest researchers working on PINNs, scientific computing + ML, optimization, etc.

**Broader Impact Concerns:**

This paper does not raise ethical concerns.

**Claims And Evidence:**

Yes

**Claims Explanation:**

Overall, the theoretical contribution is quite solid. In particular, the FEM-to-PINN analogy is well constructed. The proof that D_PINN = G (Theorem 3.6) is clean and follows directly from Taylor expansion. Theorem 3.8 on gradient correlation improvement is also correct under Assumption 3.7. The authors also provide both theoretical justification and empirical verification (Table 4, 100% positive ratio) for this assumption.

I have a few comments on evaluation / theorem framing, but should be fixable via the TMLR process since the current eval is already quite solid and comprehensive.

First, as mentioned above, the claim that gradient correlation is a "more precise criterion" than residual loss (first mentioned in the intro / abstract) is supported only visually (Figure 1). There is no systematic quantitative comparison (like correlation between gradient correlation maps and actual error maps v.s. residual loss maps and error maps across benchmarks).

Second, Theorem 3.6 is a single-step Taylor expansion showing two quantities defined to be nearly the same thing are equal. The "necessary and sufficient condition" framing overstates the mathematical contribution. I personally feel that the real novelty here is the definition and FEM interpretation instead of the proof.

**Requested Changes:**

1. Please quantitatively evaluate gradient correlation v.s. residual loss as a failure diagnostic (for example, you could compute spatial correlation or IoU with ground-truth error maps across all benchmarks) rather than relying solely on visual evidence. This makes the first contribution bullet point more solid.

2. Could you reframe Theorem 3.6 to emphasize the definitional/interpretive contribution rather than the mathematical proof, and acknowledge the circularity in the "necessary and sufficient" claim?

3. Could you investigate and discuss why Transformer-based baselines diverge on Fluid Dynamics (your hardest benchmark)? Is it inherent to attention under high-order gradients, or an implementation/hyperparameter issue? This matters for understanding the scope of ProPINN's advantage. It'd be even better if you could have more baselines work on that benchmark since making KAN work might be a bit tough.

---

> ### Author Response · Authors · 2026-05-08
>
> We sincerely thank Reviewer 6buy for the detailed and constructive comments.
>
> >**Q1:** "Please quantitatively evaluate gradient correlation v.s. residual loss as a failure diagnostic..."
>
> Thank you for this important suggestion. We have added a quantitative comparison between residual loss and low-gradient-correlation in $\underline{\text{Table 8 in Revised Paper}}$. We do not directly use point-wise correlation or IoU with the full error map because gradient correlation is a neighborhood-wise propagation quantity: it measures whether supervision can propagate across nearby regions, and is therefore more naturally interpreted as an indicator of propagation barriers rather than point-wise error magnitude.
>
> Accordingly, we adopt a boundary-localization evaluation. We define the boundary of the top-10% high-error region in the error map as the failure boundary, and compare how close the top-10% regions identified by residual loss and low-gradient-correlation are to this boundary. The results show that low-gradient-correlation localizes the failure boundary more accurately: for Vanilla PINN, the average distance decreases from 0.5047 to 0.3702, yielding a 26.64% improvement; for ProPINN, it decreases from 0.3413 to 0.2600, yielding a 23.83% improvement. These results support our claim that residual loss mainly reflects local PDE violation, while gradient correlation is more suitable for identifying propagation barriers or failure boundaries.
>
> >**Q2:** "Could you reframe Theorem 3.6 to emphasize the definitional/interpretive contribution rather than the mathematical proof, and acknowledge the circularity in the 'necessary and sufficient' claim?"
>
> Thank you for pointing this out. We agree that the previous “necessary and sufficient” wording could overstate the mathematical framing. We have therefore revised $\underline{\text{Theorem 3.6 in Revised Paper}}$ to emphasize its definitional and interpretive role. Specifically, starting from the FEM-inspired stiffness perspective, we define propagation stiffness in PINNs and show that, under $\underline{\text{Definition 3.4 in Revised Paper}}$,
> $$
> D_{\mathrm{PINN}}(\mathbf{x},\mathbf{x}') = G_{u_\theta}(\mathbf{x},\mathbf{x}').
> $$
> Thus, the main contribution is not a technically complex proof, but the connection between a physically meaningful propagation concept and a computable gradient-correlation diagnostic for PINN architectures.
>
> >**Q3:** "Could you investigate and discuss why Transformer-based baselines diverge on Fluid Dynamics (your hardest benchmark)? Is it inherent to attention under high-order gradients, or an implementation/hyperparameter issue? This matters for understanding the scope of ProPINN's advantage. It'd be even better if you could have more baselines work on that benchmark since making KAN work might be a bit tough."
>
> Thank you for this valuable suggestion. We re-trained and inspected PINNsFormer and SetPINN on the Fluid Dynamics benchmark. Both Transformer-based baselines still exhibited severe training instability: after about 50k training steps for PINNsFormer and 5k steps for SetPINN, the training loss and gradient norm rapidly increased beyond 100 and then diverged to NaN. In contrast, the non-Transformer baselines remained stable under the same task and training protocol.
>
> We think this instability is related to high-order automatic differentiation in PINN training. Fluid Dynamics involves 2D+time Navier-Stokes equations with multiple physical outputs, requiring derivatives through complex PDE residuals. Attention modules introduce more complex cross-point interactions and derivative paths, which can amplify gradients and destabilize optimization in this setting. By contrast, ProPINN uses a lightweight MLP-based multi-region mixing module, which strengthens local gradient coupling while avoiding the complicated high-order derivative paths introduced by attention. We have added this discussion to clarify the scope of ProPINN's advantage.

---

### Review · Reviewer_Frix · 2026-04-08

**Summary Of Contributions:**

This paper investigates the propagation failure phenomenon in Physics-Informed Neural Networks (PINNs), which undermines training stability. The authors attribute this failure to the conventional single-point processing architecture of PINNs, where gradient correlations among nearby collocation points remain low. To address this issue, they propose ProPINN, a method that effectively unifies the gradients of regional points to improve propagation. The effectiveness of the approach is demonstrated through empirical experiments.

**Audience:**

Yes

**Audience Explanation:**

PINN training stability is a fundamental concern in the AI for Science community.

**Claims And Evidence:**

Yes

**Claims Explanation:**

To define propagation failure in PINNs, the authors recall that such failure refers to the inability to obtain a correct solution across the entire domain. For successful training, correct supervision of model outputs must propagate from initial or boundary points to the interior domain. To analyze this failure, they formally define propagation failure in Definition 3.4, drawing on analyses from Finite Element Methods (FEMs) and establishing a connection with gradient correlation in Theorem 3.6.
To address this failure, the authors propose differential perturbation and demonstrate the resulting improvement in gradient correlation in Theorem 3.8. To validate the efficiency of the perturbation method, they conduct comparisons across various benchmark experiments.

**Requested Changes:**

1. Will the perturbation method introduce bias to PINN training? Specifically, is the optimal solution of the perturbed loss still a valid solution to the PDE?
2. How does the perturbation method scale to high-dimensional problems? Although PDE tasks are typically formulated in two or three dimensions, scalability remains an important consideration.
3. What is the impact of the perturbation range and the number of perturbations on training stability (e.g., Gradient correlation improvement) and performance (e.g., accuracy)? A more rigorous trade-off analysis (theoretical or empirical) should be conducted.
4. Is there any connection between the proposed method and adversarial training (randomized smoothing), which shares the same high-level idea of perturbation? A discussion of this relationship would be valuable.

Cohen, Jeremy, Elan Rosenfeld, and Zico Kolter. "Certified adversarial robustness via randomized smoothing." ICML. PMLR, 2019.

---

> ### Author Response · Authors · 2026-05-08
>
> We sincerely thank Reviewer Frix for the valuable and constructive feedback.
>
> >**Q1:** "Will the perturbation method introduce bias to PINN training? Specifically, is the optimal solution of the perturbed loss still a valid solution to the PDE?"
>
> Thank you for raising this important question. The differential perturbation in ProPINN does not change the PDE objective or the target solution. The PDE residual, initial condition, and boundary condition losses are still evaluated on the original collocation, initial, and boundary points, and the final prediction remains $u_\theta(\mathbf{x})$ at the original coordinate $\mathbf{x}$. The perturbed points are only used as representation-level local augmentation inside the shared projection and multi-region mixing module.
>
> Therefore, ProPINN does not optimize a different or perturbed PDE. Instead, the perturbation is designed to couple nearby gradient paths and improve local propagation. A valid solution to the original PDE remains the valid target of the optimization.
>
> >**Q2:** "How does the perturbation method scale to high-dimensional problems? Although PDE tasks are typically formulated in two or three dimensions, scalability remains an important consideration."
>
> Thank you for the insightful question. We agree that extremely high-dimensional PDEs may introduce additional computational overhead, because more augmented points may be required to cover local regions in high-dimensional spaces. We have explicitly discussed this limitation in $\underline{\text{Appendix H in Original Paper}}$.
>
> As stated there, this limitation can potentially be addressed by integrating amortization techniques with ProPINN, which can amortize the computation over the optimization process via randomization. Since this paper focuses on propagation failures rather than high-dimensional PDE solvers, we leave this as future work.
>
> >**Q3:** "What is the impact of the perturbation range and the number of perturbations on training stability (e.g., Gradient correlation improvement) and performance (e.g., accuracy)? A more rigorous trade-off analysis (theoretical or empirical) should be conducted."
>
> Thank you for the constructive suggestion. We have provided an empirical hyperparameter analysis in $\underline{\text{Appendix D in Original Paper}}$, covering the number of perturbations, perturbation region size, and number of scales. The results show a clear accuracy-efficiency trade-off: more perturbations generally improve performance by strengthening regional gradient coupling, but also increase computation; overly large perturbation regions may introduce noisy local information, especially for PDEs with sharp variations.
>
> Besides, we note that training stability and gradient correlation are related but do not determine each other in a simple monotonic way. To further clarify the diagnostic role of gradient correlation, we added $\underline{\text{Table 8 in Revised Paper}}$, where low-gradient-correlation is shown to localize propagation barriers more accurately than residual loss. This complements the hyperparameter analysis by explaining why improving regional gradient coupling is useful for propagation, even though its relationship with training loss is not necessarily monotonic.
>
> >**Q4:** "Is there any connection between the proposed method and adversarial training (randomized smoothing), which shares the same high-level idea of perturbation? A discussion of this relationship would be valuable."
>
> Thank you for this helpful suggestion. We have added $\underline{\text{Remark 3.8 in Revised Paper}}$ to discuss this connection. Although ProPINN and adversarial training both use input perturbations, their motivations are different. Adversarial training or randomized smoothing usually assumes that nearby perturbed samples preserve the same label and aims to improve robustness to input perturbations.
>
> In contrast, each spatiotemporal coordinate in PINNs corresponds to a potentially different physical state. Therefore, ProPINN does not enforce prediction robustness under perturbed coordinates. Its perturbation is better viewed as representation-level local augmentation, whose differential design aims to unite nearby region gradients and enhance local propagation. Thus, ProPINN uses perturbation to improve PINN optimization and supervision propagation, rather than to certify or enforce adversarial robustness.

---

> > ### Comment · Reviewer_Frix · 2026-06-08
> > **Response**
> >
> > I thank the authors for their detailed response and the additional empirical experiments. These have adequately addressed my comments.

---

### Review · Reviewer_EErC · 2026-04-28

**Summary Of Contributions:**

This paper addresses a fundamental bottleneck in the training of Physics-Informed Neural Networks known as propagation failure where correct boundary/initial information fails to reach the interior of the domain.

**Strengths**
- Providing formal theory of propagation failure and showing low gradient correlation as root cause
- Proposing ProPINN, a novel lightweight architecture that uses a multi-region mixing mechanism and differential perturbation to increase the correlation of gradients across a local neighborhood.
- ProPINN demonstrates notable reduction in error compared to other architectures like transformer-based models while being 2 to 3 times faster.

**Weaknesses**
- Assumptions of the specific optimizer: definition of $D_{PINN}$ relies on updates in the direction of gradient. For optimizers like Adam or L-BFGS, this definition may not accurately capture the "stiffness" during the updates from these optimizers.
- Vagueness in some statements, like "small correlation" in Thm. 3.6 . Without a way to derive the threshold for smallness from the PDE itself, this criterion is somehow heuristic.
- Success of the model is tied to additional hyper-parameters, region-sizes and number of perturbations, which can depend on the PDE being solved.

Overall, the paper makes strong contributions toward fundamental architectural understanding of why PINNs fail and how to fix it

**Audience:**

Yes

**Audience Explanation:**

The paper addresses a fundamental issue if PINNs, provides evidence why the training of PINNs may fail, and proposed a novel and efficient architecture.

**Claims And Evidence:**

Yes

**Claims Explanation:**

The evidence presented in the paper is generally accurate, clear, and highly convincing. The proofs to the theoretical results are correct. Moreover, the authors provide multiple experiment setups to show the architectural efficiency and performance of ProPINN compared to other state-of-the-art models.

**Requested Changes:**

**Theoretical findings**

1- Theorem 3.6 can explicitly state the numerical equivalence between the physical stiffness $D_{PINN}$ and the gradient correlation $G$. Currently, this equivalence is buried in the proof.
2- Explicitly State Assumptions: The authors should clarify that the derivation of $D_{PINN}$ assumes a standard gradient step $\theta - \lambda \round u / \round \theta$. To address the use of complex optimizers like L-BFGS or Adam, the paper can discuss why gradient correlation remains the "root cause" even when update directions deviate from the raw gradient.
3- While the paper provides separate plots for gradient correlation and training loss, a unified plot showing the rate of convergence as a function of the Minimum Gradient Correlation would provide direct empirical evidence of their findings.

---

> ### Author Response · Authors · 2026-05-08
>
> We sincerely thank Reviewer EErC for the detailed and constructive review.
>
> >**Q1:** "Theorem 3.6 can explicitly state the numerical equivalence between the physical stiffness $D_{\mathrm{PINN}}$ and the gradient correlation $G$. Currently, this equivalence is buried in the proof."
>
> Thank you for pointing this out. We have revised $\underline{\text{Theorem 3.6 in Revised Paper}}$ to explicitly state the numerical equivalence between the PINN stiffness coefficient and gradient correlation:
> $$
> D_{\mathrm{PINN}}(\mathbf{x},\mathbf{x}')=G_{u_\theta}(\mathbf{x},\mathbf{x}').
> $$
> Thus, this key equivalence is now stated in the theorem itself, rather than only in the proof as in the original paper.
>
> >**Q2:** "Explicitly State Assumptions: The authors should clarify that the derivation of $D_{\mathrm{PINN}}$ assumes a standard gradient step."
>
> Thank you for the helpful suggestion. We have clarified this in $\underline{\text{Definition 3.4 in Revised Paper}}$: $D_{\mathrm{PINN}}$ measures the effect of updating the PINN with an infinitesimal standard gradient step at $\mathbf{x}$. Our derivation analyzes intrinsic local coupling induced by model gradients, rather than claiming that every practical optimizer exactly follows raw gradients.
>
> >**Q3:** "To address the use of complex optimizers like L-BFGS or Adam, the paper can discuss why gradient correlation remains the 'root cause' even when update directions deviate from the raw gradient."
>
> We appreciate this important comment. Although Adam and L-BFGS modify raw gradients through adaptive scaling, preconditioning, or historical curvature information, their update directions are still gradient-induced. Thus, if the parameter gradients from two nearby collocation points are nearly orthogonal, an update driven by one point still has weak local influence on the other. This weak architectural coupling is what gradient correlation captures.
>
> We added $\underline{\text{Remark 3.7 in Revised Paper}}$ to clarify this optimizer-agnostic interpretation. Empirically, $\underline{\text{Table 6 in Original Paper}}$ reports Adam+L-BFGS results where ProPINN remains consistently better than all baselines, supporting that our observation is not tied to a single optimizer.
>
> >**Q4:** "Vagueness in some statements, like 'small correlation' in Thm. 3.6. Without a way to derive the threshold for smallness from the PDE itself, this criterion is somehow heuristic."
>
> Thank you for raising this concern. We agree that the threshold should not be interpreted as a universal PDE-derived constant. The $\epsilon$ is an empirical diagnostic threshold. Our main theoretical contribution is to transform the physically motivated PINN stiffness coefficient into a computable gradient-correlation quantity. The exact threshold may depend on the PDE, model scale, and numerical precision; we have revised the wording to avoid overstating the universality of "smallness."
>
> >**Q5:** "While the paper provides separate plots for gradient correlation and training loss, a unified plot showing the rate of convergence as a function of the Minimum Gradient Correlation would provide direct empirical evidence of their findings."
>
> Thank you for the suggestion. We agree that connecting optimization behavior with minimum gradient correlation is valuable. However, training loss is not always a reliable convergence indicator for propagation failure. As shown in $\underline{\text{Figure 9 in Original Paper}}$, PINNs may reach low residual loss while still having large test error, because interior regions do not receive correct propagated supervision. Improving propagation can even temporarily slow residual-loss decrease, since the model no longer collapses to a trivial low-residual solution.
>
> >**Q6:** "Success of the model is tied to additional hyper-parameters, region-sizes and number of perturbations, which can depend on the PDE being solved."
>
> We agree that region sizes and perturbation numbers are additional hyperparameters. We provided a detailed analysis in $\underline{\text{Appendix D in Original Paper}}$ on perturbation number, region size, and number of scales. The results show a clear trade-off: more perturbations strengthen regional gradient coupling but increase computation, while overly large regions may introduce noise for PDEs with sharp variations.
>
> We also note that PDE solvers commonly require PDE-dependent hyperparameters, and strong baselines such as PINNsFormer also involve architecture and sampling choices. To reduce sensitivity to one region size, ProPINN adopts a multi-scale design and uses the same default configuration across all tested PDEs.

---

> > ### Comment · Reviewer_EErC · 2026-05-11
> > **thanks for the responses and revision**
> >
> > I appreciate the authors' responses and the revised version of the paper. They have addressed my concerns and comments

---

### Decision · Action_Editor_Ft8t · 2026-07-09

**Recommendation:** Accept as is

**Audience:**

Yes

**Audience Explanation:**

The contributions of this paper make the work relevant to researchers in machine learning, scientific computing, and neural PDE solvers.

**Claims And Evidence:**

Yes

**Claims Explanation:**

The paper provides a solid theoretical analysis and comprehensive empirical validation of propagation failure in PINNs. The authors establish a meaningful connection between FEM-inspired propagation stiffness and gradient correlation, and propose ProPINN as an effective architectural solution. The theoretical claims are supported by formal analysis, while extensive experiments across diverse PDE benchmarks demonstrate consistent improvements over existing PINN architectures. The authors have also addressed the reviewers' concerns regarding optimizer assumptions, diagnostic evaluation, Transformer-based baselines, and hyperparameter sensitivity through additional analysis and clarifications in the revision.